

# Doping lattice non-Abelian quantum Hall states

Zhengyan Darius Shi[1⋆], Carolyn Zhang[2†] and Senthil Todadri[1‡]

**1** Department of Physics, Massachusetts Institute of Technology,
Cambridge, Massachusetts 02139, USA
**2** Department of Physics, Harvard University, Cambridge, Massachusetts 02138, USA

⋆ zdshi@mit.edu , † cczhang@fas.harvard.edu , ‡ senthil@mit.edu

## Abstract

We study quantum phases of a fluid of mobile charged non-Abelian anyons, which arise upon doping the lattice Moore-Read quantum Hall state at lattice filling $\nu = 1/2$ and its generalizations to the Read-Rezayi ($RR_k$) sequence at $\nu = k/(k+2)$. In contrast to their Abelian counterparts, non-Abelian anyons present unique challenges due to their non-invertible fusion rules and non-Abelian braiding structures. We address these challenges using a Chern-Simons-Ginzburg-Landau (CSGL) framework that incorporates the crucial effect of energy splitting between different anyon fusion channels at nonzero dopant density. For the Moore-Read state, we show that doping the charge $e/4$ non-abelion naturally leads to a fully gapped charge-2 superconductor without any coexisting topological order. The chiral central charge of the superconductor depends on details of the interactions determining the splitting of anyon fusion channels. For general $RR_k$ states, our analysis of states obtained by doping the basic non-abelion $a_0$ with charge $e/(k+2)$ reveals a striking even/odd pattern in the Read-Rezayi index $k$. We develop a general physical picture for anyon-driven superconductivity based on charge-flux unbinding, and show how it relates to the CSGL description of doped Abelian quantum Hall states. Finally, as a bonus, we use the CSGL formalism to describe transitions between the $RR_k$ state and a trivial period-$(k+2)$ CDW insulator at fixed filling, driven by the gap closure of the fundamental non-Abelian anyon $a_0$. Notably, for $k = 2$, this predicts a period-4 CDW neighboring the Moore-Read state at half-filling, offering a potential explanation of recent numerical observations in models of twisted $MoTe_2$.

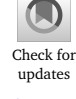

# 1   Introduction

The experimental discovery [1–6] of fractional quantum anomalous Hall (FQAH) phases in two-dimensional Van der Waals materials inspires novel theoretical questions. In contrast to quantum Hall phases in a fractionally filled Landau level, where single-electron motion is confined into cyclotron orbits by a large magnetic field, charged excitations in FQAH phases occupy Bloch states with well-defined crystal momenta and non-trivial dispersion. Doping the FQAH insulator therefore induces a finite density of mobile charged anyons, providing a physical realization of the "anyon fluid" first conceptualized in the pioneering work of Laughlin [7]. Even in the dilute limit, the long-range statistical interactions between arbitrarily distant anyons make the problem intrinsically interacting and prevent any "free-particle" description. Understanding the novel itinerant phases that emerge in such anyonic quantum matter presents a fascinating theoretical challenge.

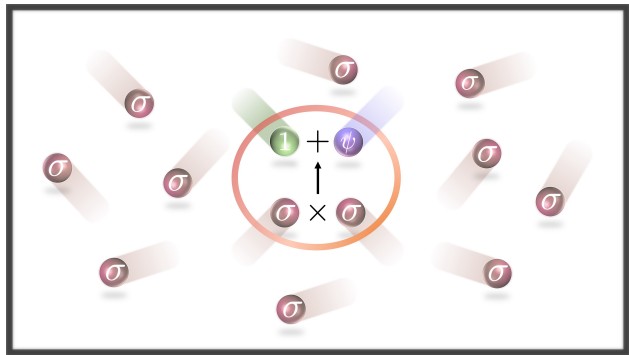

Figure 1: An itinerant fluid of non-Abelian $\sigma$ anyons that satisfy the simple fusion rule $\sigma \times \sigma = 1 + \psi$. The robust topological degeneracy of non-Abelian anyons at large spatial separation is split by interactions at short distances. We will study the charged non-abelion fluid obtained by doping the closely related Moore-Read state (and their generalizations). The splitting between fusion channels 1 and $\psi$ plays a crucial role in our analysis.

In a recent work, two of us analyzed the landscape of itinerant phases that emerge in doped FQAH insulators with Abelian anyons, finding both chiral topological superconductors and charge-ordered Fermi liquids [8].[1] Remarkably, a subsequent experiment [11] in twisted MoTe$_2$ indeed finds a superconducting state in the vicinity of the Jain state at lattice filling $\nu = 2/3$, in sharp contrast to the persistent plateaus in conventional quantum Hall systems with large external magnetic fields. The central goal of this work is to generalize the constructions in Ref. [8] to doped non-Abelian states. Although non-Abelian states at zero external magnetic field have not been observed in experiments thus far, numerical studies provide encouraging evidence for their existence in models of a variety of moire materials [12–17]. This tantalizing possibility motivates us to search for novel itinerant phases in doped non-Abelian states and make concrete physical predictions for near-term experiments.

A key physical property that distinguishes non-Abelian topological orders from their Abelian counterpart is the existence of non-invertible fusion between non-Abelian anyons. In an Abelian topological order, a pair of anyons $a$ and $b$ always fuse into a unique third anyon $c$. The operation of fusion therefore endows the set of Abelian anyons with the mathematical structure of a group. In contrast, non-Abelian topological orders contain at least one pair of anyons $a$ and $b$ with multiple fusion outcomes. This fusion structure is captured by an equation $a \times b = \sum_c N^c_{ab} c$, where the non-negative integer coefficients $N^c_{ab}$ are the *fusion multiplicities* and the different $c$'s for which $N^c_{ab} \neq 0$ are the distinct *fusion channels*. Physically, the presence of multiple fusion channels implies that the Hilbert space associated with multiple isolated non-Abelian anyons is not one-dimensional. Moreover, the energies of different states in this Hilbert space are exactly degenerate when anyons are infinitely far apart from each other. This topological degeneracy, robust against all local perturbations to the Hamiltonian, is a hallmark feature of non-Abelian topological order.

In systems with a finite density of non-Abelian anyons, this topological degeneracy immediately breaks down. As the typical inter-anyon spacing becomes finite, interactions between the anyons generically lift the degeneracy completely. Any theory for a doped non-Abelian FQAH insulator must therefore simultaneously keep track of the near degeneracy at large anyon separation and the splitting of distinct fusion channels at short distances. The fate of the doped non-Abelian state depends on the interplay between these two effects, posing a novel conceptual challenge beyond its Abelian analog.

---

[1]See Refs. [9, 10] for related works in different physical settings.

A powerful framework for addressing this challenge is the effective Chern-Simons-Ginzburg-Landau (CSGL) theory for non-Abelian quantum Hall states. Since the pioneering works of Refs. [18–22], it has been widely appreciated that the topological orders of Abelian fractional quantum Hall phases can be described by Chern-Simons (CS) theories with $U(1)$ gauge fields. More recently, these constructions have been generalized to many non-Abelian fractional quantum Hall states, with $U(1)$ gauge fields replaced by non-Abelian gauge fields [23–25]. While pure CS theories have no finite-energy excitations, gapped dispersive anyons can be introduced through dynamical matter fields that couple to the CS theories. When the CS theory has a global $U(1)$ symmetry, the gapped anyons are endowed with fractional $U(1)$ charge as well as non-Abelian fusion and braiding. Furthermore, as we explain in Section 2.2, lattice translation symmetry enrichment can be incorporated by introducing multiple species of matter fields that transform projectively under translation action. Such $U(1)$ + lattice translation symmetry-enriched CS-matter theories provide a faithful description of the universal low energy properties of non-Abelian FQAH phases.

In this work, we will focus on the itinerant phases that arise from doping a sequence of fermionic Read-Rezayi states $RR_k$ at lattice filling $\nu = k/(k+2) \mod \mathbb{Z}$, where $k$ is a positive integer. Note that there is an additional integer parameter $M$ in the labeling of generalized Read-Rezayi states at filling $\nu = k/(kM+2)$. We will specialize to the case $M = 1$, although our framework easily generalizes to other values of $M$. The $M = 1$ family includes the famous Moore-Read state [26, 27] ($k = 2$) as well as the $\mathbb{Z}_3$ parafermion state [28] ($k = 3$), which are promising candidates for the quantum Hall states realized at Landau level filling $\nu = 5/2$ and $\nu = 12/5$ in the conventional large-field FQH setting [29–31].[2]

The doping process introduces a finite density of the cheapest charged quasiparticle. We will for the most part assume that this excitation has the minimum fractional charge $e/(k+2)$. In $RR_k$ with $k > 3$, there are multiple anyons with this minimum charge. We will focus on the basic non-Abelian anyon[3] $a_0$ with this minimum charge which in addition has the minimum topological spin $1/(2k+4)$. A simplifying feature shared by all the $RR_k$ states is that pair fusion of $a_0$ only produces two fusion channels $b_s$ and $b_t$, which we refer to as the singlet and triplet respectively. The meaning of this nomenclature will become clear after we introduce the field theory for $RR_k$ in Section 2.

Following Ref. [23], we describe this family of states using a unified $U(2)$ CSGL theory with $k$-dependent CS levels (this formalism will be reviewed in Section 2). Our analysis of the CSGL theories at finite charge density reveals an intriguing even/odd pattern in the Read-Rezayi index $k$. When $k$ is even, the doped FQAH states always become superconducting. The exact nature of the superconductor depends on the energy splitting of the two fusion channels $b_s$ and $b_t$. If the singlet is favored, we find a charge-2 topological superconductor with chiral central charge $c_- = -1/2$ for all values of $k$. If the triplet is favored, we instead obtain a topologically trivial charge-$k$ superconductor with $c_- = 0$. When $k$ is odd, the situation is drastically different. If the singlet channel is favored, we again obtain a charge-2 topological superconductor with $c_- = -1/2$, which is in the same phase as the superconductor obtained in the even-$k$ scenario. On the other hand, if the triplet channel is favored, the system realizes an intermediate-temperature non-Fermi liquid phase with a strong pairing instability. The low-temperature paired state is an ordinary Fermi liquid made up of charge-$k$ fermions that coexists with a period-$(k + 2)$ charge density wave (CDW) order. This last possibility is particularly striking: despite the presence of charge-1 electrons in the microscopic system, correlations intrinsic to the Read-Rezayi state enforce a clustering of electrons into charge-$k$ composites in the doped theory. This clustering phenomenon is reminiscent of the original intuition for the

---

[2]At $\nu = 12/5$, the candidate state is actually the particle-hole conjugate of $RR_3$.

[3]We will also briefly describe the alternate possibility, available for odd $k$, when the doped quasiparticle is an Abelian anyon with the minimum charge $e/(k+2)$.

formation of Read-Rezayi states in Ref. [28] and warrants a deeper analysis in the future.

As a bonus, we also use the CSGL theory to describe a phase transition (continuous at the mean-field level) between the $RR_k$ state and a topologically trivial CDW insulator with period $(k + 2)$. This transition occurs when the energy gap of the basic non-Abelian anyon $a_0$ closes continuously as a function of some tuning parameter at fixed lattice filling. Specializing to $k = 2$, our construction provides a possible explanation for the previously mysterious period-4 CDW that was found in the vicinity of a lattice Moore-Read state through numerical studies of models of twisted $MoTe_2$ [32].

The context and results in this paper should be distinguished from Ref. [33], which explored the physics of doped non-Abelian spin liquids within a field theoretic model. There the doped anyons are electrons bound to a neutral non-Abelian anyon and the statistical interactions between the anyons induce an effective attraction in the BCS channel. This BCS attraction leads to a direct condensation of Cooper pairs which leaves the parent non-Abelian topological order intact. This physics is qualitatively different from the superconductors (and other phases) found in our work.

The rest of this paper is organized as follows. In Section 2, we review the construction of the lattice Moore-Read state, which is the simplest non-Abelian FQAH phase. We will begin with an algebraic description of the Moore-Read topological order and its enrichment by $U(1)$ and lattice translation symmetries in Section 2.1. This will be followed by a Chern-Simons-Ginzburg-Landau (CSGL) description of the same state in Section 2.2. The CSGL Lagrangian provides a powerful framework for analyzing phase transitions out of the FQAH state and provides important intuition for the anyon dynamics, which we discuss in Sections 2.3 and 2.4. In Section 3, we outline our strategy for analyzing the doped CSGL theory and then present a derivation of the two superconducting phases that emerge upon doping the lattice Moore-Read state. This analysis generalizes in a straightforward way to all the Read-Rezayi states, which we treat in Section 4. Following the field-theoretic derivations, we develop a physical interpretation of the superconductors in terms of charge-flux unbinding in Section 5. We conclude in Section 6 with a discussion on experimental signatures (including an associated anomalous vortex glass phase in the presence of disorder), important open questions, and future directions.

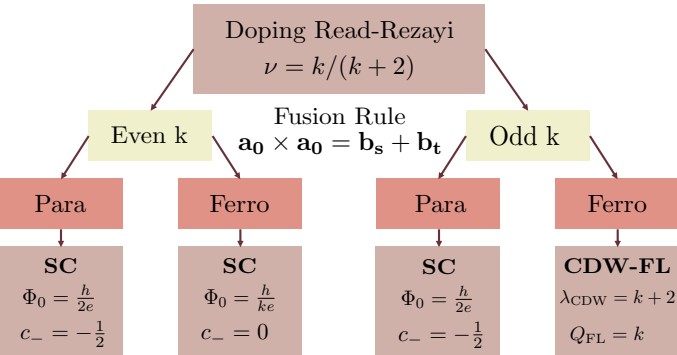

Figure 2: A summary of our results for doping the minimally charged non-Abelian anyon $a_0$ in Read-Rezayi states. Within the $U(2)$ CSGL theory, the itinerant phase realized by the system depends on whether the $U(2)$ fluxes vanish on average ("paramagnetic") or polarize in some direction ("ferromagnetic"). These two cases correspond to the energetic preference of $b_s$ or $b_t$ in the pairwise fusion of $a_0$. Each possible phase is labeled by a set of universal data: $\Phi_0$ is the flux quantization, $c_-$ is the chiral central charge, $\lambda_{\mathrm{CDW}}$ is the period of the CDW (in units of the lattice spacing) and $Q_{\mathrm{FL}}$ is the electric charge of the fermions that make up the FL.

## 2 Construction of non-Abelian FQAH phases

### 2.1 Lattice Moore-Read state: Anyons and symmetry enrichment

In this section, we review the construction of lattice Moore-Read states realized at filling $\nu = 1/2 \mod \mathbb{Z}$. The topological order associated with the Moore-Read state is $[\text{Ising} \times U(1)_8]/\mathbb{Z}_2$. The Ising sector has anyons $1, \sigma, \psi$ and the $U(1)_8$ theory has anyons $a^n$ for $n = 0, \ldots, 7$. The modding by $\mathbb{Z}_2$ means that $(1, \psi)$ are glued to $a^n$ with $n$ even, and $\sigma$ is glued to $a^n$ with $n$ odd.[4]

A topological order is uniquely specified by a set of anyon types, their fusion rules, $R$ and $F$ symbols, and a chiral central charge (for a review, see Ref. [34]). In our discussion, we will only need information about fusion rules and topological spin (determined by the $R$ symbol). In the $U(1)_8$ sector, the topological spin of the anyon $a^n$ is $h_{a^n} = n^2/16$, and fusion is determined by a simple multiplicative rule $a^n \times a^m = a^{n+m}$. In the Ising sector, the non-trivial topological spins are $h_\sigma = 1/16$ and $h_\psi = 1/2$, and we have a more complicated fusion structure

$$\sigma \times \sigma = 1 + \psi, \quad \psi \times \psi = 1, \quad \psi \times \sigma = \sigma \times \psi = \sigma. \tag{1}$$

Combining the fusion rules and topological spins in the two sectors gives the corresponding data in the product theory Ising $\times U(1)_8$. The $\mathbb{Z}_2$ quotient eliminates some anyons but does not modify the fusion and braiding structure of the remaining anyons.

In a lattice electronic system that realizes the Moore-Read topological order, the global symmetries of interest are $U(1)$ and lattice translations. The global $U(1)$ charge assignment is such that $a^n$ has charge $ne/4$, where $e$ is the electron charge. The lattice translation action on anyons is determined by filling constraints [35–39]. At half-filling, we must place at each unit cell a background Abelian anyon that carries charge $e/2$. For the lattice Moore-Read state, there are two possible choices, $\nu = a^2$ and $\nu = a^2\psi$, which lead to distinct symmetry-enriched topological orders.[5] Since these two choices give identical phases at nonzero doping, we will fix $\nu = a^2$ for concreteness. In the presence of this background vison, translations generally fail to commute when acting on a single anyon excitation $b$

$$T_x T_y T_x^{-1} T_y^{-1} [b] = e^{i\theta_{\nu b}} [b], \tag{2}$$

where $\theta_{\nu b} = 2\pi p/q$ (with $p, q$ coprime) is the braiding phase between $b$ and $\nu$. When $q > 1$, this action is projective and the $b$ anyon carries well-defined momenta in a $q$-fold reduced Brillouin zone. Within the reduced Brillouin zone, the band structure of $b$ can be described by $q$ degenerate species related to each other through the action of lattice translations (see Ref. [8] for a more detailed review).

### 2.2 Chern-Simons-Ginzburg-Landau description of the lattice Moore-Read state

The algebraic description of the lattice Moore-Read state presented so far efficiently captures universal properties associated with the topological order. However, since we will be interested in the dynamics of itinerant anyons, it is useful to have an alternative field-theoretic description. While Abelian topological orders can be captured by a finite collection of $U(1)$ gauge fields through the powerful $K$-matrix formalism [22], non-Abelian topological orders generally require the introduction of non-Abelian gauge fields. For the Moore-Read state (and

---

[4]This corresponds to gauging the 1-form symmetry generated by $\psi a^4$ (attached to the physical fermion). The resulting theory consists of anyons that braid trivially with $\psi a^4$.

[5]Note that if the UV charge filling is $1/2 \mod 1$ instead of $1/2$ exactly, then $a^6$ and $a^6\psi$ can also be chosen as the vison. But these are related to $a^2\psi$ and $a^2$ via the attachment of a microscopic electron $\psi a^4$ that braids trivially with all other anyons. As a result, these choices do not give new symmetry enriched topological orders.

more generally the Read-Rezayi states), the most convenient effective field theory description involves a single $U(2)$ gauge field [23] (see Refs. [40–42] for alternative gauge theory descriptions based on various parton constructions). Dynamical dispersive anyon excitations can be included by coupling matter fields to the $U(2)$ gauge theory. The total Lagrangian that includes both gauge and matter sectors will be referred to as the Chern-Simons-Ginzburg-Landau (CSGL) Lagrangian. In what follows, we will construct the CSGL Lagrangian for the Moore-Read state and describe how to extract properties of the Moore-Read topological order from the field-theoretic perspective. To streamline the discussion, we will state some technical results without proof and refer to more careful derivations in Appendix A.

The Lie group $U(2)$ can be decomposed as $U(2) = [SU(2) \times U(1)]/\mathbb{Z}_2$ and the Lie algebra of $U(2)$ has a three-dimensional $SU(2)$ component and a one-dimensional $U(1)$ component. The most general Chern-Simons theory for a $U(2)$ gauge field therefore involves a pair of Chern-Simons levels $k_1$ and $k_2$ for the $SU(2)$ and $U(1)$ sectors respectively. Such a theory will be referred to as $U(2)_{k_1,k_2}$. The Lagrangian for a pure $U(2)_{k_1,k_2}$ Chern-Simons theory in the absence of dynamical matter fields takes the form

$$L = -\frac{k_1}{4\pi} \mathrm{Tr}\left( c \wedge dc + \frac{2}{3} c \wedge c \wedge c \right) + \frac{2k_1 - k_2}{4} \frac{1}{4\pi} \mathrm{Tr}\, c \wedge d\, \mathrm{Tr}\, c \,, \tag{3}$$

where $c$ is a $U(2)$ gauge field and $\wedge$ is the standard wedge product acting on the spacetime indices of $c$. For notational convenience, we will drop the $\wedge$ symbols from now on.

By decomposing $c$ into its $SU(2)$ and $U(1)$ components, one can verify that $k_1, k_2$ indeed correspond to the $SU(2)$ and $U(1)$ Chern-Simons levels, which are quantized to be integers. As explained in Appendix A, the $\mathbb{Z}_2$ quotient imposes an additional condition $2k_1 + k_2 \in 4\mathbb{Z}$. This Chern-Simons theory contains a local fermion in its spectrum[6] if and only if $2k_1 + k_2 = 4$ mod 8; otherwise, it is bosonic. Since we will be interested in the fermionic Moore-Read and Read-Rezayi states, this additional quantization condition must be satisfied.

Anyons in this Chern-Simons theory are represented as endpoints of Wilson line operators in various irreducible representations of the $U(2)$ gauge field. These representations are labeled by the half-integer-quantized $SU(2)$ spin $j$ and the integer-quantized $U(1)$ charge $n$, subject to the constraint $j + n/2 \in \mathbb{Z}$. By the standard Chern-Simons quantization rules [43], the non-redundant anyons of the $U(2)_{k_1,k_2}$ theory satisfy $j \leq |k_1|/2, n < |k_2|$ and have topological spin

$$h^{(k_1,k_2)}_{(j,n)} = \frac{j(j+1)}{k_1 + 2} + \frac{n^2}{2k_2} \,. \tag{4}$$

The fusion rules between distinct anyons match the fusion rules of $SU(2)$ and $U(1)$ representations, modulo a truncation on the maximum allowed $SU(2)$ and $U(1)$ representations:

$$(j_1, n_1) \times (j_2, n_2) = \sum_{j=|j_1-j_2|}^{\min(|k_1|/2, |k_1|-j_1-j_2)} (j, [n_1 + n_2 \mod |k_2|]) \,. \tag{5}$$

The Moore-Read topological order corresponds to a $U(2)_{k_1,k_2} \times U(1)_1$ theory with $k_1 = 2$ and $k_2 = -8$ (which indeed satisfies the quantization condition $2k_1 + k_2 = 4 \mod 8$). The decoupled $U(1)_1$ sector is just a fermionic integer quantum Hall state which is stacked with the $U(2)_{k_1,k_2}$, and is necessary for obtaining the correct $c_- = 3/2$. The Moore-Read Lagrangian therefore takes the form

$$L_{\mathrm{MR}} = -\frac{2}{4\pi} \mathrm{Tr}\left( c\, dc + \frac{2}{3} c^3 \right) + \frac{3}{4\pi} \mathrm{Tr}\, c\, d\, \mathrm{Tr}\, c - \frac{1}{4\pi} \alpha d\alpha \,, \tag{6}$$

---

[6]This kind of Chern-Simons theory is referred to as a spin Chern-Simons theory in the field theory literature.

where $c$ is a $U(2)$ gauge field and $\alpha$ is a $U(1)$ gauge field. Within this construction, the vacuum maps to $(0, 0)$, the basic non-Abelian anyon $\sigma a$ maps to $(1/2, 1)$, while the fermion $\psi$ maps to $(1, 0)$. The fusion rules of these basic anyons in the $U(2)_{2,-8}$ theory can be worked out explicitly:

$$(1/2, 1) \times (1/2, 1) = (0, 2) + (1, 2), \quad (1/2, 1) \times (1, 0) = (1/2, 1), \quad (1, 0) \times (1, 0) = (0, 0). \quad (7)$$

One can verify that these rules are in exact agreement with the fusion rules described in Section 2.1. Note that $(0, 2)$ and $(1, 2)$ can be identified with $b_s$ and $b_t$ in Section 1. This identification justifies the singlet/triplet notation, as $(0, 2)$ and $(1, 2)$ transforms in the singlet/triplet representations of $SU(2)$ respectively.

While the $U(2)_{2,-8} \times U(1)_1$ Chern-Simons theory provides a correct description of the Moore-Read topological order, its abstract nature obscures the physical origin of the Moore-Read state. To fill in this gap, we present in Appendix A.3 an explicit derivation of the $U(2)_{2,-8} \times U(1)_1$ Chern-Simons theory from $p + ip$ pairing of composite fermions. This derivation allows us to make an explicit identification of $\sigma a$ and $\psi$ with the $h/(2e)$ vortices and Bogoliubov quasiparticles of the $p + ip$ superconductor.

Having identified the correct Chern-Simons theory for the Moore-Read topological order, we are now ready to implement the $U(1)$ and lattice translation symmetry enrichment. The $U(2)_{2,-8} \times U(1)_1$ Chern-Simons theory itself enjoys a $U(1) \times U(1)$ symmetry associated with the conservation of flux for $\alpha$ and $\operatorname{Tr} c$. The physical $U(1)$ charge conservation symmetry in the microscopic system corresponds to a diagonal $U(1)$ subgroup of this $U(1) \times U(1)$ global symmetry. Therefore, we can keep track of the symmetry enrichment through a mutual Chern-Simons term that couples the external electromagnetic gauge field $A$ (technically a $\text{spin}_{\mathbb{C}}$ connection, as explained in Ref. [44]) to $(\alpha - \operatorname{Tr} c)$. The $U(1)$-enriched Lagrangian then takes the form

$$L_{\text{MR}}[A] = -\frac{2}{4\pi} \operatorname{Tr}\left(c \, dc + \frac{2}{3} c^3\right) + \frac{3}{4\pi} \operatorname{Tr} c \, d \operatorname{Tr} c - \frac{1}{4\pi} \alpha d\alpha + \frac{1}{2\pi} (\alpha - \operatorname{Tr} c) dA. \quad (8)$$

The coupling to $A$ endows the anyon $(j, n)$ with a physical electric charge $ne/4$. This charge assignment correctly captures the $e/4$ charge of the basic non-Abelian anyon $\sigma a = (1/2, 1)$ as well as the neutral fermion $\psi = (1, 0)$.

The pure Chern-Simons theory in (8) assigns an infinite energy gap to all the anyon excitations. To describe dynamical anyon excitations, we must introduce matter fields in various irreducible representations of $U(2)$. For the basic non-Abelian anyon $(1/2, 1)$, the corresponding matter field lives in the fundamental representation of $U(2)$. Furthermore, the general arguments in Section 2.1 imply that lattice translations act projectively on the non-Abelian anyon as

$$T_x T_y T_x^{-1} T_y^{-1} [\sigma a] = e^{2\pi i/4} \sigma a. \quad (9)$$

As a result, $\sigma a$ carries well-defined crystal momenta in a 4-fold reduced Brillouin zone and the band structure of $\sigma a$ can be described by 4 degenerate species related to each other through the action of lattice translations. This constraint forces the low-energy Lagrangian to contain four species of matter fields $\Phi_\alpha$ each transforming in the fundamental representation of $U(2)$:

$$L_{\text{MR}} = -\frac{2}{4\pi} \operatorname{Tr}\left(c \, dc + \frac{2}{3} c^3\right) + \frac{3}{4\pi} \operatorname{Tr} c \, d \operatorname{Tr} c - \frac{1}{4\pi} \alpha d\alpha + \frac{1}{2\pi} (\alpha - \operatorname{Tr} c) dA + \sum_{\alpha=1}^{4} L[\Phi_\alpha, c], \quad (10)$$

where the gauge-invariant matter field Lagrangian takes the general form

$$L[\Phi_\alpha, c] = |D_\mu \Phi_\alpha|^2 - m^2 |\Phi_\alpha|^2 - V |\Phi_\alpha|^4 + \dots, \quad (11)$$

and the $\Phi_\alpha$ fields (with index $\alpha$ defined mod 4) transform under lattice translations as

$$T_x : \Phi_\alpha \to \Phi_{\alpha+1}, \quad T_y : \Phi_\alpha \to e^{2\pi i(\alpha+1)/4} \Phi_{\alpha+1}. \quad (12)$$

The Lagrangian in (10) provides a complete description of the lattice Moore-Read state, enriched with both $U(1)$ and lattice translation symmetries. From now on, we will refer to (10) as the CSGL Lagrangian of the Moore-Read state. The CSGL Lagrangian of the $RR_k$ Read-Rezayi states can be constructed through a similar procedure and is described in Appendix A. Doping the system (i.e. adding a chemical potential) modifies the above Lagrangian with an additional term linear in $\Phi_\alpha^* i D_0 \Phi_\alpha$, which we suppress until analyzing the doped theory in Sections 3 and 4.

Finally, let us briefly comment on the set of gauge-invariant local operators in the CSGL theory, organized by the charge that they carry under the external gauge field $A$. The basic matter field that explicitly enters the Lagrangian is $\Phi_\alpha$, which carries the $(1/2, 1)$ representation. Algebraically, we expect all local excitations appearing in fusion powers of $(1/2, 1)$ and its anti-particle to survive as *finite-energy* excitations of the CSGL theory. How do we explicitly construct the local operators that create these excitations?

The simplest local operators are charge-neutral quasiparticle/quasihole pairs made up of equal numbers of $\Phi_\alpha^*$ and $\Phi_\alpha$ fields. Examples include $\Phi_\alpha^* \Phi_\beta$ and polynomials built out of them. The projective action of lattice translations implies that these operators carry nonzero lattice momentum when $\alpha \neq \beta$ and overlap with the microscopic charge density wave order parameters. Another class of charge-neutral local operators are gauge-invariant combinations of the $U(2)$ field strength tensors such as $\operatorname{Tr} F$ and $\operatorname{Tr} F_{\mu\nu} F^{\mu\nu}$. Correlation functions of these operators in the CSGL theory encode the spectrum of low-energy neutral excitations such as the magnetoroton.

The local charged excitations in the CSGL theory must carry integer charge (the fractionally charged anyons are attached to Wilson lines and cannot be local). Due to the mutual Chern-Simons term between $A$ and $\operatorname{Tr} c$, these charged operators are monopole creation/annihilation operators that insert/remove fluxes of $c$, which we construct explicitly in Appendix C. In particular, the microscopic electron/Cooper pair embeds in the low-energy theory as a monopole that inserts $2\pi/4\pi$ flux of $\operatorname{Tr} c$ respectively. Importantly, all local charged excitations have a finite energy gap, which can in principle be determined by computing monopole correlation functions in the appropriate charge sector.

## 2.3 Phase transitions out of the Moore-Read state at fixed lattice filling

Before moving on to the effects of doping, we comment on a simple application of the CSGL Lagrangian in (10) to the competition between distinct correlated phases at fixed lattice filling $\nu = 1/2$. In ordinary Ginzburg-Landau theories with a symmetry group $G$, matter fields live in representations of $G$ and various condensation pathways of the matter fields describe phase transitions between a disordered phase and ordered phases in which $G$ is spontaneously broken to a subgroup. In CSGL theories, the matter fields transform in representations of the gauge group $G$. Condensing the matter field Higgses the gauge group $G$ to a smaller subgroup and describes a phase transition into a distinct topological order.

Using the CSGL description of the Moore-Read state, we can immediately identify different Higgsing patterns that describe phase transitions out of the Moore-Read state at fixed lattice filling. Microscopically, these phase transitions can be driven by tuning various parameters in the experimental system such as the displacement field and the strength of the moire potential. A Higgs transition in which a matter field in the $(j, n)$ representation is "condensing" corresponds to a physical situation in which tuning a parameter closes the energy gap of the $(j, n)$ anyon and all of its fusion products while preserving the gaps of the remaining anyons.

The simplest case is where the elementary non-Abelian anyon $\sigma a \equiv (1/2, 1)$ closes its gap across the phase transition. Since every other anyon can be generated from $(1/2, 1)$ through fusion, such a transition would destroy the entire non-Abelian topological order. The resulting phase can be derived from the CSGL Lagrangian. Condensing the $(1/2, 1)$ matter

field $\Phi_\alpha$ Higgses $U(2)$ down to a $U(1)$ subgroup generated by the lower-diagonal component $c \to \mathrm{diag}(0, c_\downarrow)$. After Higgsing, the Lagrangian transforms to

$$L_{\mathrm{MR}} \to \frac{1}{4\pi} c_\downarrow dc_\downarrow - \frac{1}{4\pi} \alpha d\alpha + \frac{1}{2\pi}(\alpha - c_\downarrow)dA, \tag{13}$$

which indeed describes a trivial topological phase with chiral central charge $c_- = 0$. Moreover, since $\Phi_\alpha$ itself transforms under the lattice translation symmetry, the condensation of $\Phi_\alpha$ spontaneously breaks the lattice translation symmetry, inducing a period-4 charge density wave. We have therefore found a continuum field-theory description of a direct transition between the Moore-Read state and a trivial period-4 CDW insulator at fixed lattice filling. When generalized to lattice filling $\nu = k/(k+2)$, the same construction describes a transition between the Read-Rezayi $\mathrm{RR}_k$ state and a period-$(k+2)$ CDW insulator.

This family of transitions is remarkable for two different reasons. First, an analogous transition between an Abelian Jain state at filling $\nu = p/(2p+1)$ and a trivial CDW insulator has no known field-theory description in general [45]. It is therefore surprising that a description exists when we replace Abelian Jain states with much more complicated non-Abelian states. Second, when $k$ is even, although the CDW insulator is topologically trivial, its period is twice as large as the minimal period $(k+2)/2$ that is compatible with filling constraints at $\nu = k/(k+2)$. Remarkably, a recent numerical study of twisted $\mathrm{MoTe}_2$ at half-filling of the second miniband indeed finds a topologically trivial period-4 CDW in close competition with the Moore-Read state [32]. The existence of a natural path from $\mathrm{RR}_k$ to this large-period CDW may be related to specific electronic correlations that stabilize the parent $\mathrm{RR}_k$ topological order in the first place.

## 2.4 Doping lattice non-Abelian states: General strategy

Having reviewed the algebraic and field-theoretic constructions of the lattice non-Abelian states, we are now prepared to study the effects of doping. In the Abelian case, a simple heuristic for deducing the nature of the doped state has been developed using the invertible fusion structure of Abelian anyons [8]. As explained in Section 1, such an invertible fusion breaks down for anyons in non-Abelian topological phases. In this section, our goal is to develop a refined heuristic for the doped non-Abelian state by incorporating the splitting of degeneracies between different fusion channels. The conclusions that we draw will provide an intuitive framework for understanding the formal field-theoretic results in Section 3 and Section 4.

We begin by reviewing the Abelian heuristic. Near the simplest Jain state at lattice filling $\nu = 2/3$, the nature of the doped state depends on the energetics of different anyon species in the parent FQAH insulator [8]. If the cheapest anyon has charge $q = \pm 2/3$, then the ground state is a topological superconductor with 4 neutral Majorana edge modes; if the cheapest anyon has charge $q = \pm 1/3$, the ground state is an ordinary Fermi liquid with a doping-induced period-3 charge density wave. The dichotomy between these two scenarios can be understood through a simple heuristic. Suppose that the cheapest anyon $a_0$ carries charge $Q(a_0)$ and that the cheapest *local excitation* constructed from fusion products of $a_0$ is $a_0^k$ with integer charge $p = kQ(a_0)$. When $p$ is odd, this local excitation is a fermion moving in a dispersive band and a compressible metallic phase with a Fermi surface naturally emerges at generic lattice filling. When $p$ is even, this local excitation $a_0^k$ is instead bosonic and prefers to condense at generic lattice filling, giving rise to a charge-$p$ superconductor. When applied to the Jain states at $\nu = p/(2p+1)$, this heuristic agrees with the field-theoretic analysis, although it cannot resolve the finer structures of the metallic/superconducting phases such as concomitant symmetry-breaking and/or residual topological order.

Using the intuition outlined in the previous paragraph, we can attempt a naive generalization to the non-Abelian context. Suppose that the cheapest charged anyon $a_0$ is a non-Abelian anyon with charge $Q(a_0)$. Fusion products of $a_0$ with itself generally produce a sum over multiple fusion channels. Now let $k$ be the smallest integer such that $a_0^k$ contains a local excitation. If we denote this local excitation by $l$, then this condition implies the following fusion equation

$$a_0^k = n_l \, l + \dots, \tag{14}$$

where $n_l \geq 1$ is the fusion multiplicity of $l$ and $\dots$ denotes a sum over non-trivial anyons. The total electric charge of $l$ is $p = kQ(a_0)$. When $p$ is odd, $l$ is a local charge-$p$ fermion and a metallic state with a Fermi surface of $l$ naturally emerges. When $p$ is even, the basic local excitation is the charge-$p$ boson $l$ and all local fermions are absent in the low-energy spectrum. The natural doped state is therefore a charge-$p$ superconductor. In both cases, the metallic/superconducting state could coexist with a residual topological order formed by anyons that braid trivially with $a_0$.

This naive generalization fails to account for the splitting of degeneracies between different fusion channels. Take for example the lattice Moore-Read state. If we choose $a_0$ to be the basic non-Abelian anyon $\sigma a = (1/2, 1)$ with physical charge $Q(\sigma a) = e/4$, then the smallest integer $k$ for which $(\sigma a)^k$ contains a local excitation is $k = 4$ and the corresponding fusion relation is

$$(\sigma a)^4 \equiv (1/2, 1)^4 = [(0, 2) + (1, 2)] \times [(0, 2) + (1, 2)] = 2(0, 4) + 2(1, 4). \tag{15}$$

While $(1, 4)$ and $(0, 4)$ both carry unit electric charge, $(0, 4)$ is an anyon while $(1, 4)$ is the microscopic electron. If $(0, 4)$ has a lower energy gap than $(1, 4)$, then an electronic Fermi surface is unlikely to form. Therefore, the appearance of a particular local fermion in $a_0^k$ does not guarantee the emergence of a metallic phase in the doped state. Indeed, our field-theoretic analysis in Section 3 reveals that a superconductor always emerges upon doping the Moore-Read state *despite the presence of a local fermion in the low-energy spectrum*.[7]

The lesson we learn is that the splitting of fusion channels at nonzero dopant density plays a crucial role in determining the structure of low-energy local excitations and the fate of the doped non-Abelian state. In the CSGL description, this splitting is related to the interactions between anyons mediated by the non-Abelian gauge field. The different choices of preferred fusion channels lead to distinct Higgsing patterns for the non-Abelian gauge fields and therefore distinct itinerant phases. A detailed analysis of these different possibilities is the subject of the next two sections.

## 3  Doping the lattice Moore-Read state

In this section, we analyze the doped lattice Moore-Read state from a field-theoretic perspective. At very low dopant density, disorder and/or long-range Coulomb interactions tend to localize the anyons, leading to the familiar quantum Hall plateau. However, when the dopant density is sufficiently large, the kinetic energy of the dispersive anyons in a Chern band can overcome localization and induce an itinerant anyonic fluid. In this regime, we can use the continuum CSGL Lagrangian (10) which incorporates both the parent topological order and its enrichment with $U(1)$ and lattice translation symmetries. Since the $U(1)$ gauge field $\alpha$ is at level 1 (i.e. it describes an integer quantum Hall state), we can integrate it out and generate

---

[7]Although this result about the doped many-body phase cannot be rigorously attributed to the energetic preference of $(0, 4)$ over $(1, 4)$, the general phenomenon of fusion channel splitting does provide valuable intuition, as we discuss in more detail throughout Section 3 and Section 4.

a simpler description of the same state

$$L_{\text{MR}} = -\frac{2}{4\pi} \text{Tr}\left( c\, dc + \frac{2}{3} c^3 \right) + \frac{3}{4\pi} \text{Tr}\, c\, d\, \text{Tr}\, c - \frac{1}{2\pi} \text{Tr}\, c\, dA + \text{CS}[A, g] + \sum_{\alpha=1}^{4} L[\Phi_\alpha, c], \quad (16)$$

where the background Chern-Simons term $\text{CS}[A, g]$ is defined as

$$\text{CS}[A, g] = \frac{1}{4\pi} A dA + 2\text{CS}_g. \quad (17)$$

In writing down the integer quantum Hall response $\text{CS}[A, g]$, we have implicitly placed the theory on a closed, oriented spacetime manifold of the form $M_3 = M_2 \times M_1$ (where $M_2$ is the two-dimensional spatial manifold and $M_1$ is the one-dimensional temporal manifold) equipped with a metric $g$. The gravitational Chern-Simons term $2\text{CS}_g$ encodes the quantized thermal Hall conductance carried by the integer quantum Hall edge, which is twice as large as the contribution from a single Majorana edge mode.

In the undoped theory, $\Phi_\alpha$ is gapped and nucleates the $e/4$ non-abelion. To analyze the doped theory, it is convenient to decompose the $U(2)$ gauge field $c$ as $c = \sum_{a=0}^{3} c_a T^a$ where $T^0 \equiv I$ and $T^a \equiv \sigma^a$ are the $SU(2)$ generators for $a \neq 0$. Similarly, we will define the non-Abelian gauge current $J_\Phi = \sum_{a=0}^{3} J_\Phi^a T^a / 2$ such that the coupling between $\Phi_\alpha$ and $c$ can be written as

$$\text{Tr}\, J_\Phi c = \frac{1}{2} \sum_{a,b=0}^{3} c_b J_\Phi^a \text{Tr}\left[ T^a T^b \right] = \sum_{a=0}^{3} c_a J_\Phi^a. \quad (18)$$

With this notation, the equations of motion for $c_a$ take the form

$$\frac{8}{2\pi} dc_0 + \star J_\Phi^0 + \frac{2}{2\pi} dA = 0, \quad -\frac{2}{2\pi} F^a + \star J_\Phi^a = 0, \quad F = dc + c^2. \quad (19)$$

From the Lagrangian, we know that the physical charge density that couples to the temporal component of the external gauge field $A$ is the Abelian flux $\frac{2}{2\pi} \nabla \times c_0$. At a nonzero dopant density, the Abelian flux develops a nonzero expectation value which induces a non-zero Abelian matter density $J_\Phi^0$ through the Chern-Simons coupling. However, fixing the dopant density does not fix the non-Abelian fluxes $F^a$ and leaves the non-Abelian matter density $J_\Phi^a$ unconstrained as well.

What is the physical information encoded in the different choices of non-Abelian fluxes and non-Abelian matter densities? In the CSGL description, a single $(1/2, 1)$ anyon is sourced by the $\Phi_\alpha$ fields, which live in the $j = 1/2$ representation of $SU(2)$. Following the algebraic discussion in Section 2.4, the fusion of two $\Phi_\alpha$ decomposes into a singlet channel $(0, 2)$ and a triplet channel $(1, 2)$, whose energy gaps are generically split by interactions. The Abelian/non-Abelian matter density $J_\Phi^0/J_\Phi^a$ therefore maps to the density of fusion products in the $(0, 2)/(1, 2)$ channel respectively. If pair fusion favors $(0, 2)$ energetically, the triplet matter density vanishes on average and the entire $U(2)$ gauge field remains alive. On the other hand, if pair fusion favors $(1, 2)$, the triplet matter density turns on. Without loss of generality, we can choose the matter density to be polarized in the $\sigma^3$ direction of $SU(2)$ color space. Turning on an expectation value for $J_\Phi^3$ Higgses $U(2)$ down to the Abelian $U(1) \times U(1)$ subgroup. Drawing an analogy with $U(2)$-symmetric magnets, we refer to the $U(2)$-preserving situation as "paramagnetic" and the $U(2)$-Higgsed situation as "ferromagnetic".

Which of these possibilities is favored in a particular realization of the lattice Moore-Read state? In general, this is a dynamical question that is difficult to resolve using effective field theory. However, a reasonable estimate can be made in the weak-coupling regime of the CSGL theory. To access the weak-coupling regime, we introduce a Yang-Mills term for the non-Abelian gauge field on top of the Chern-Simons term. The total Lagrangian then takes the

form

$$L_{\text{MR}} = \frac{1}{2g^2} \operatorname{Tr} F \wedge \star F - \frac{2}{4\pi} \operatorname{Tr} \left( c \, dc + \frac{2}{3} c^3 \right) + \frac{3}{4\pi} \operatorname{Tr} c \, d \operatorname{Tr} c$$
$$- \frac{1}{2\pi} \operatorname{Tr} c \, dA + \operatorname{CS}[A, g] + \sum_{\alpha=1}^{4} L[\Phi_\alpha, c]. \tag{20}$$

In 2+1D, the Yang-Mills coupling $g^2$ and the mass parameter $M$ appearing in the matter field Lagrangian both have mass dimension 1. Working in a regime where $g^2 \ll M$, we can treat localized $\Phi_\alpha$ excitations as probe fields whose effective interactions are determined by integrating out the massive gauge fluctuations. This calculation is carried out explicitly for an arbitrary $U(2)_{k_1,k_2}$ gauge theory in Appendix B. Specializing to the Moore-Read state, we find that the effective interactions between a pair of $(1/2, 1)$ anyons receive separate contributions from the $U(1)$ and $SU(2)$ sectors. In the $U(1)$ sector, the interaction is always repulsive at long distances and decays exponentially with a characteristic length $\xi_{U(1)} = \frac{2\pi}{g^2|k_2|}$. In the $SU(2)$ sector, the interaction remains repulsive in the triplet sector but becomes attractive in the singlet sector with a different decay length $\xi_{SU(2)} = \frac{2\pi}{g^2|k_1|}$. As a result, in the weak-coupling limit, the singlet fusion channel $(0, 2)$ generically has a lower energy than the triplet fusion channel $(1, 2)$, thereby favoring the "paramagnetic" case. As the coupling $g^2$ increases, we enter a strong-coupling regime where the effective interactions are more difficult to determine. In principle, there could exist a critical $g_c$ (depending on $k_1$) above which the $(1, 2)$ channel has a lower energy. If so, the "ferromagnetic" situation becomes favored at $g > g_c$ and an entirely different itinerant phase emerges.

While this two-particle calculation is instructive, we caution that the extrapolation from two-particle energetics to many-body ground states involves a leap of faith. Drawing an analogy with the BCS theory of superconductivity in normal metals, the two-particle calculation is similar to the calculation of phonon-induced attraction between pairs of isolated electrons. Although this attraction motivates the condensation of Cooper pairs, rigorously establishing superconductivity requires a definitive solution of the many-body problem. Therefore, the intuition we gain from the two-particle calculation should only serve as a heuristic guide.

With this discussion in mind, it is reasonable to expect that both the "paramagnetic" and "ferromagnetic" cases can be realized in different regions of the phase diagram of a doped lattice Moore-Read state. Moreover, a transition between these two cases may be possible as microscopic couplings are tuned to vary the ratio between the effective Maxwell coupling $g^2$ and the mass parameter $M$. In the subsequent sections, we will consider these two cases in turn and determine universal properties of the resulting itinerant phases.

### 3.1  $U(2)$ color paramagnet: Chiral topological superconductor

We first assume that in the doped state, the singlet channel $(0, 2)$ is favored over the triplet channel $(1, 2)$. In this case, the full $U(2)$ gauge structure of the Lagrangian stays un-Higgsed. At the mean-field level, the non-Abelian fluxes $F^a$ as well as the non-Abelian matter densities $J_\Phi^a$ have vanishing expectation values. In the absence of any external magnetic field, the Abelian sector of (19) reduces to

$$-\frac{8}{2\pi} \langle \nabla \times \mathbf{c_0} \rangle = \langle \rho_\Phi^0 \rangle, \tag{21}$$

where $\rho_\Phi^0$ is the temporal component of the Abelian current $J_\Phi^0$. It follows that the $\Phi$ are at a net Landau level filling $-8$, and each of the 4 species is at a filling $-2$. At this filling, we consider the simplest gapped state in which each boson species $\Phi_\alpha$ forms a bosonic integer quantum Hall state (i.e. an SPT protected by a $U(2)$ global symmetry) [46]. This phase has indeed been observed over a range of interaction parameters in numerical studies of two-component boson systems subject to orbital magnetic fields [47]. For each boson species, the $U(1)$ Hall

conductivity is then quantized to $-2$ while the $SU(2)$ Hall conductivity is quantized to $+1$. To match this response, the $U(2)$ Chern-Simons term generated by integrating out each $\Phi_\alpha$ must take the form

$$L[\Phi_\alpha, c] \to \frac{1}{4\pi} \operatorname{Tr}\left( cdc + \frac{2}{3}c^3 \right) - \frac{1}{4\pi} \operatorname{Tr} c \, d \operatorname{Tr} c \,. \tag{22}$$

Since the four boson species go into the same mean-field state, we can multiply the Chern-Simons term above by 4 and combine with the terms already present in (A.11) to get

$$L_{\text{eff}} = \frac{2}{4\pi} \operatorname{Tr}\left( cdc + \frac{2}{3}c^3 \right) - \frac{1}{4\pi} \operatorname{Tr} c \, d \operatorname{Tr} c + \frac{1}{2\pi} Ad \operatorname{Tr} c + \text{CS}[A, g] \,. \tag{23}$$

By matching with the general form in Appendix A, we see that this final Lagrangian describes a $U(2)_{-2,0} \times U(1)_1$ Chern-Simons theory. Since the Abelian sector has level 0, the $U(1)$ gauge field $c_0$ couples only to the external gauge field $A$ with coefficient 2. The fluctuations of $c_0$ therefore produces a Meissner effect for $2A$, giving rise to a charge-2 superconductor. In the Abelian sector, it appears that we have a $SU(2)_{-2}$ theory with non-trivial topological order. However, the non-abelion of the $SU(2)_{-2}$ theory, which corresponds to the $j = \frac{1}{2}$ representation, is glued to objects with odd charge under $c_0$. But these are precisely the $nh/2e$ vortices of the superconductor with $n$ odd. Thus we conclude that the non-abelion is bound to these vortices and is not an independent excitation. The quasiparticle with $j = 1$ then just becomes the Bogoliubov quasiparticle.

To complete the identification, we note that the chiral central charge has a contribution $-\frac{3}{2}$ from the $SU(2)_{-2}$ sector and $+1$ from the $\text{CS}[A, g]$ term. Thus we end up with a net chiral central charge $c_- = -\frac{1}{2}$. We conclude that the resulting state is a $p + ip$ BCS superconductor with a single chiral Majorana edge state that is counter-propagating relative to the direction of the chiral edge modes of the parent Moore-Read state. Remarkably, despite the exotic starting point of a non-Abelian anyon fluid, the final state is smoothly connected to a BCS superconductor! Of course, the mechanism to reach this state is strikingly non-BCS with no normal state Fermi surface at temperatures above the superconducting transition temperature $T_c$.

## 3.2 $U(2)$ color ferromagnet: Trivial BCS superconductor

Now we assume that in the doped theory, each $\Phi_\alpha = (\Phi_{\alpha,\uparrow}, \Phi_{\alpha,\downarrow})^T$ orders "ferromagnetically" in the $SU(2)$ color space, thereby Higgsing the $U(2)$ gauge group down to $U(1) \times U(1)$. Without loss of generality, we can choose a mean-field in which $\langle \Phi_\alpha^\dagger \sigma \Phi_\alpha \rangle = m\hat{z}$. At low energies, we can then restrict the $U(2)$ gauge field to $c = \operatorname{diag}(c_\uparrow, c_\downarrow)$ where $c_{\uparrow,\downarrow}$ are $U(1)$ gauge fields with ordinary $2\pi$-flux quantization. Assuming complete color polarization, we put $\Phi_{\alpha,\uparrow}$ at finite density and $\Phi_{\alpha,\downarrow}$ at zero density. Substituting the diagonal $c$ in (16) and integrating out the gapped $\Phi_{\alpha,\downarrow}$ gives

$$L = -\frac{2}{4\pi}\left( c_\uparrow dc_\uparrow + c_\downarrow dc_\downarrow \right) + \frac{3}{4\pi}\left( c_\uparrow + c_\downarrow \right) d\left( c_\uparrow + c_\downarrow \right) + \frac{1}{2\pi} Ad\left( c_\uparrow + c_\downarrow \right) + \text{CS}[A, g] + \sum_{\alpha=1}^{4} L[\Phi_{\alpha,\uparrow}, c_\uparrow]$$

$$= \frac{1}{4\pi}\left( c_\uparrow dc_\uparrow + c_\downarrow dc_\downarrow \right) + \frac{3}{2\pi} c_\uparrow dc_\downarrow + \frac{1}{2\pi} Ad\left( c_\uparrow + c_\downarrow \right) + \text{CS}[A, g] + \sum_{\alpha=1}^{4} L[\Phi_{\alpha,\uparrow}, c_\uparrow] \,. \tag{24}$$

Now the $c_\downarrow$ field does not couple to any dynamical matter and has a self Chern-Simons term at level-1. Therefore, we can integrate out $c_\downarrow$ to get

$$L = -\frac{8}{4\pi} c_\uparrow dc_\uparrow - \frac{2}{2\pi} Adc_\uparrow + \sum_{\alpha=1}^{4} L[\Phi_{\alpha,\uparrow}, c_\uparrow] \,. \tag{25}$$

It follows from the equation of motion for $c_\uparrow$ that each of the 4 species of $\Phi_{\alpha,\uparrow}$ is at a Landau level filling 2. Putting these into boson integer quantum hall states, we see that the CS terms for $c_\uparrow$ cancel and we are left with

$$L = -\frac{2}{2\pi} A d c_\uparrow. \tag{26}$$

This describes a trivial charge-2 BCS superconductor with no neutral chiral edge modes. Note that in the Lagrangian of (25) a *single* $\Phi_{\alpha,\uparrow}$ particle nucleates an Abelian anyon with self-statistics $\pi/8$ and a physical electric charge $e/4$. Thus in this "color ferromagnet" state, the original non-Abelian anyon at zero density has transmuted, at finite density, into an Abelian anyon.

## 3.3 Pair-binding of non-Abelian anyons: Other chiral superconductors

In both the "paramagnetic" and "ferromagnetic" situations, we assumed that at all wavevectors,

$$2\Delta_{(1/2,1)} < \Delta_{(0,2)}, \Delta_{(1,2)}, \tag{27}$$

where $\Delta_{(j,n)}$ is the energy gap of the $(j,n)$ anyon. This assumption implies that pairs of $(1/2,1)$ anyons do not have a binding instability and the CSGL Lagrangian written in terms of matter fields in the $(1/2,1)$ representation is the correct starting point for analyzing the doped state.

It is interesting to contemplate an alternative situation in which $2\Delta_{(1/2,1)}$ is much greater than $\Delta_{(0,2)}/\Delta_{(1,2)}$ and isolated non-Abelian anyons prefer to bind into one of the two channels. In this "pair-binding" limit, single $(1/2,1)$ excitatons can be projected out of the deep IR spectrum and it is more appropriate to write down a different CSGL Lagrangian in which the only matter field lives in the $(0,2)/(1,2)$ representation of $U(2)$. What is the fate of the doped state in these two cases?

We first consider the case where pair-binding favors the $(0,2)$ anyon, which is the anti-semion. Since the anti-semion has a $\pi$ braiding phase with the vison, translation symmetry fractionalization gives two degenerate species of scalar matter fields $\Phi_\alpha$. The total effective Lagrangian then takes the form

$$L = -\frac{2}{4\pi} \text{Tr}\left( c d c + \frac{2}{3} c^3 \right) + \frac{3}{4\pi} \text{Tr} c \, d \, \text{Tr} c + \frac{1}{2\pi} \text{Tr} c \, d A + \sum_{\alpha=1}^{2} L[\Phi_\alpha, \text{Tr} c] + \text{CS}[A, g]. \tag{28}$$

The equations of motion for the Abelian sector impose the constraint

$$\sum_{\alpha=1}^{2} \rho_{\Phi_\alpha} = -\frac{2}{2\pi} \nabla \times \text{Tr} c. \tag{29}$$

Therefore, the two-component boson $\Phi_\alpha$ is at Landau level filling $-2$ with respect to the emergent magnetic field it sees. Putting this pair of bosons in the standard bosonic IQH state produces the topological response

$$L = -\frac{2}{4\pi} \text{Tr}\left( c d c + \frac{2}{3} c^3 \right) + \frac{1}{4\pi} \text{Tr} c \, d \, \text{Tr} c + \frac{1}{2\pi} \text{Tr} c \, d A + \text{CS}[A, g]. \tag{30}$$

This Lagrangian describes a $U(2)_{2,0} \times U(1)_1$ theory with chiral central charge $c_- = 5/2$, which is a topological superconductor with $\nu_{\text{eff}} = 5$ in the Kitaev classification (i.e. 5 copies of $p + ip$) [34].

Next, we consider the case where pair-binding favors the semion $(1,2)$. Since $(1,2)$ can be regarded as the bound state of $(0,-2)$ with the microscopic electron $(1,4)$, the CSGL Lagrangian now involves two degenerate species of fermionic matter fields $\psi_\alpha$, each carrying

charge $-1$ under $\operatorname{Tr} c$ and charge 1 under the $\text{spin}_{\mathbb{C}}$ connection $A$

$$L = -\frac{2}{4\pi}\operatorname{Tr}\left(cdc + \frac{2}{3}c^3\right) + \frac{3}{4\pi}\operatorname{Tr} c\, d\operatorname{Tr} c + \frac{1}{2\pi}\operatorname{Tr} c\, dA + \sum_{\alpha=1}^{2} L[\psi_\alpha, A - \operatorname{Tr} c] + \text{CS}[A, g]. \quad (31)$$

The equations of motion for $\operatorname{Tr} c$ place the fermions $\psi_\alpha$ at Landau level filling $-2$ with respect to the emergent magnetic field it sees. Putting $\psi_\alpha$ in the $\nu = -2$ fermionic IQH state gives the effective Lagrangian

$$L = -\frac{2}{4\pi}\operatorname{Tr}\left(cdc + \frac{2}{3}c^3\right) + \frac{1}{4\pi}\operatorname{Tr} c\, d\operatorname{Tr} c - \frac{1}{2\pi}\operatorname{Tr} c\, dA - \text{CS}[A, g]. \quad (32)$$

This Lagrangian describes a $U(2)_{2,0} \times U(1)_{-1}$ theory with chiral central charge $c_- = 1/2$, which maps to the standard $p + ip$ superconductor.

This latter result admits a simple interpretation from the perspective of composite fermion constructions. Recall that in the gauge-invariant version of the standard composite fermion construction (see Appendix A.3), we represent the microscopic electron $c$ as $c = f\phi$ where $\phi$ is a boson and $f$ is a fermion. At filling $\nu = 1/2$, we can put $\phi$ in the $U(1)_2$ bosonic Laughlin state and $f$ in the $p + ip$ superconductor state to construct the Moore-Read state. The semion $(1, 2)$ in the Moore-Read state corresponds precisely to the semion of the $U(1)_2$ state formed by $\phi$. Doping the semion in the $U(1)_2$ state produces a trivial superfluid in which $\phi$ condenses, thereby identifying $f$ with the physical electron $c$. Therefore, the effect of doping is to "undo" the composite fermion flux attachment and transform a $p + ip$ superconductor of composite fermions into a $p + ip$ superconductor of physical electrons.

# 4  Doping the lattice Read-Rezayi states

The field-theoretic construction in Section 3 naturally generalizes to a larger family of Read-Rezayi topological orders $\text{RR}_k$ that can appear at lattice filling $\nu = k/(k+2)$. The Chern-Simons field theory description of $\text{RR}_k$ is $U(2)_{k,-2(k+2)} \times U(1)_1$. Note that $k = 1$ and $k = 2$ reduce to the $\nu = 1/3$ Laughlin state and the Moore-Read state respectively. The filling constraint is satisfied by placing a background vison $v$ in each unit cell of the lattice, which is nucleated by an adiabatic $2\pi$ flux insertion. Since the Hall conductivity is $\sigma_H = k/(k+2)$, the background anyon $v$ carries charge $ke/(k+2)$ and self-statistics $\pi k/(k+2)$. The braiding phase of an arbitrary anyon $b$ around $v$ is given by $2\pi Q(b)$ where $Q(b)$ is the electric charge of $b$.

The Lagrangian for the TQFT of the Read-Rezayi state (see Appendix A) is

$$L_{\text{RR}_k}[c] = -\frac{k}{4\pi}\operatorname{Tr}\left(cdc + \frac{2}{3}c^3\right) + \frac{k+1}{4\pi}\operatorname{Tr} c\, d\operatorname{Tr} c - \frac{1}{2\pi}Ad\operatorname{Tr} c + \text{CS}[A, g]. \quad (33)$$

The distinct anyons transform in $U(2)$ representations labeled by $(j, n)$ with $j = 0, \frac{1}{2}, 1, \cdots \frac{k}{2}$, $n = 0, 1, \cdots |2(k + 2)| - 1$ with the further restriction that $j + n/2 \in \mathbb{Z}$. As in the Moore-Read case, we will assume that the cheapest charged excitation is an anyon $a_0$ with minimal fractional charge $Q = e/(k+2)$. For general $k > 2$, there are multiple choices of $a_0$, labeled by $(j, 1)$ with $j \in \mathbb{Z} + 1/2$, which carry this minimal charge. In Section 4.1, we will focus on the case where $a_0$ is the minimally charged non-Abelian anyon with $j = 1/2$. We will comment on other choices in Section 4.2 and analyze the simplest examples with $j = k/2$ and odd $k$. For odd $k$, $(k/2, 1)$ is always an Abelian anyon and the properties of the doped state can be easily deduced.

## 4.1 Doping the minimally charged non-Abelian anyon

In this section, we assume that the cheapest charged excitation is $a_0 \equiv (1/2, 1)$, so that the doped system can be modeled as a finite-density fluid of $a_0$. Using the general formula in (2), the electric charge of $a_0$ fixes the projective translation action:

$$T_x T_y T_x^{-1} T_y^{-1}[a_0] = e^{2\pi i/(k+2)}[a_0]. \tag{34}$$

It follows that $a_0$ moves in a $(k+2)$-fold enlarged unit cell, and can be described in terms of $k+2$ species that are related by the action of lattice translations.

To describe the doped system, we closely follow the analysis of Section 3 and introduce $k+2$ bosonic fields $\Phi_\alpha$, each transforming in the fundamental representation of $U(2)$:

$$L = L_{\mathrm{RR}_k}[c] + \sum_{\alpha=1}^{k+2} L[\Phi_\alpha, c]. \tag{35}$$

The equations of motion for $c_a$ now take the form

$$\frac{2(k+2)}{2\pi} dc_0 + J_\Phi^0 = 0, \quad -\frac{k}{2\pi} F^a + J_\Phi^a = 0. \tag{36}$$

At a nonzero dopant density, the Abelian flux is required to develop a nonzero expectation value which induces a non-zero Abelian matter density $J_\Phi^0$. However, the non-Abelian flux $F^a$ and the non-Abelian matter density $J_\Phi^a$ are unconstrained and must be determined dynamically. Like in the Moore-Read case, the fusion of $a_0 \equiv (1/2, 1)$ with itself produces two distinct fusion products $(0, 2)$ and $(1, 2)$, which correspond to the singlet and triplet channels of $SU(2)$. When the singlet channel $(0, 2)$ is energetically favored, $F^a$ and $J_\Phi^a$ vanish on average and the $U(2)$ gauge group remains un-Higgsed. When the triplet channel $(1, 2)$ is favored instead, $F^a$ and $J_\Phi^a$ develop an expectation value at the mean-field level, thereby Higgsing $U(2)$ down to $U(1) \times U(1)$. In complete analogy with the Moore-Read case, we refer to these situations as "paramagnetic" and "ferromagnetic" respectively.

In the "paramagnetic" case, each boson species $\Phi_\alpha$ only sees a background Abelian magnetic field $\frac{1}{2\pi} \nabla \times c_0$. The Abelian equation of motion in (36) fixes the Landau level filling of each boson species to be $-2$. As in Section 3.1, we put each of these in the bosonic IQH state and get the effective Lagrangian

$$L = \frac{2}{4\pi} \operatorname{Tr}\left(cdc + \frac{2}{3}c^3\right) - \frac{1}{4\pi} \operatorname{Tr} c \, d \operatorname{Tr} c + \frac{1}{2\pi} Ad \operatorname{Tr} c + \mathrm{CS}[A, g]. \tag{37}$$

Remarkably, all factors of $k$ cancel out. The effective Lagrangian is the same as for the doped Moore-Read state (23) and describes a chiral topological BCS superconductor with a single counter-propagating Majorana edge mode.

In the "ferromagnetic" situation, the fate of the doped theory depends crucially on the value of $k \bmod 2$. As in Section 3.2, we focus on the case with complete $SU(2)$ color polarization so that the $\Phi_{\alpha,\downarrow}$ field is at zero density for all $\alpha$. Restricting the $U(2)$ gauge field to $c = \operatorname{diag}(c_\uparrow, c_\downarrow)$, we can rewrite (35) as

$$L = -\frac{k}{4\pi}(c_\uparrow dc_\uparrow + c_\downarrow dc_\downarrow) + \frac{k+1}{4\pi}(c_\uparrow + c_\downarrow)d(c_\uparrow + c_\downarrow) + \frac{1}{2\pi} Ad(c_\uparrow + c_\downarrow) + \mathrm{CS}[A, g] + \sum_{\alpha=1}^{k+2} L[\Phi_{\alpha,\uparrow}, c_\uparrow]$$

$$= \frac{1}{4\pi}(c_\uparrow dc_\uparrow + c_\downarrow dc_\downarrow) + \frac{1}{2\pi} Adc_\uparrow + \sum_{\alpha=1}^{k+2} L[\Phi_{\alpha,\uparrow}, c_\uparrow] + \mathrm{CS}[A, g] + \frac{1}{2\pi}\big[A + (k+1)c_\uparrow\big]dc_\downarrow. \tag{38}$$

Since the self Chern-Simons term for $c_\downarrow$ has level-1 and $c_\downarrow$ does not couple to any matter fields, we can integrate out $c_\downarrow$ and obtain

$$
\begin{aligned}
L &= \frac{1-(k+1)^2}{4\pi}c_\uparrow dc_\uparrow + \frac{1-(k+1)}{2\pi}Adc_\uparrow + \sum_{\alpha=1}^{k+2} L[\Phi_{\alpha,\uparrow}, c_\uparrow] \\
&= -\frac{k(k+2)}{4\pi}c_\uparrow dc_\uparrow - \frac{k}{2\pi}Adc_\uparrow + \sum_{\alpha=1}^{k+2} L[\Phi_{\alpha,\uparrow}, c_\uparrow].
\end{aligned}
\tag{39}
$$

The equation of motion for $c_\uparrow$ now enforces

$$
\frac{k(k+2)}{2\pi}dc_\uparrow = J_\Phi^0.
\tag{40}
$$

Therefore, at nonzero dopant density, we have $(k+2)$ species of bosons $\Phi_\alpha$, each at Landau level filling $k$.

When $k$ is even, the simplest invertible phase that can be formed by $\Phi_\alpha$ is $k/2$ copies of the boson IQH state. When this is the case, integrating out $\Phi_{\alpha,\uparrow}$ generates a Chern-Simons term for $c_\uparrow$ at level $k(k+2)$ which cancels the background Chern-Simons term at level $-k(k+2)$. The resulting effective Lagrangian is

$$
L_{\text{even}-k} = -\frac{k}{2\pi}Adc_\uparrow,
\tag{41}
$$

which describes a trivial charge-$k$ superconductor with chiral central charge $c_- = 0$.

When $k$ is odd, a fraction of $\Phi_\alpha$ at filling $k-1$ can form $(k-1)/2$ copies of the boson IQH state. The remaining fraction at filling 1 forms a bosonic composite Fermi liquid. The total Lagrangian is therefore

$$
\begin{aligned}
L_{\text{odd}-k} &= -\frac{k(k+2)}{4\pi}c_\uparrow dc_\uparrow + \frac{(k-1)(k+2)}{4\pi}c_\uparrow dc_\uparrow - \frac{k}{2\pi}Adc_\uparrow + \sum_{i=1}^{k+2} L_{\text{CFL}}[f_i, a_i, c_\uparrow] \\
&= -\frac{k+2}{4\pi}c_\uparrow dc_\uparrow - \frac{k}{2\pi}Adc_\uparrow + \sum_{i=1}^{k+2} L_{\text{CFL}}[f_i, a_i, c_\uparrow],
\end{aligned}
\tag{42}
$$

where the bosonic CFL Lagrangian takes the standard form

$$
L_{\text{CFL}}[f_i, a_i, c_\uparrow] = L_{\text{FL}}[f_i, a_i] + \frac{1}{4\pi}(c_\uparrow - a_i)d(c_\uparrow - a_i) + 2\text{CS}_g.
\tag{43}
$$

The $k+2$ copies of bosonic CFLs interact strongly with each other through their common coupling to $c_\uparrow$. The structure of interactions between the bosonic CFLs is identical to the structure of interactions between fermionic CFLs that appear in the analysis of doped Jain states. At intermediate temperatures, this is a non-Fermi liquid metal whose detailed transport and thermodynamic properties have been worked out in Ref. [8].

At low temperature, the gauge fluctuations of $a_i$ and $c_\uparrow$ strongly enhance interspecies BCS pairing between $f_i$ and $f_j$ with $i \neq j$ [8]. In Appendix D, we work out the nature of the paired state. While the pairing between two fermionic CFLs leads to an exciton condensate, the pairing between two bosonic CFLs generates a bosonic IQH state. Moreover, due to the fractionalization of translation symmetry, the interspecies Cooper pair $f_i f_j$ carries nonzero lattice momentum. Therefore, in addition to generating an integer quantum Hall response, the paired state spontaneously breaks the lattice translation symmetry, giving rise to a period-$(k+2)$ charge density wave.

Having understood a pair of bosonic CFLs, let us return to the Lagrangian in (42). Since $k+2$ is odd, we can only pair $k+1$ species of the CFLs and leave one species alone. The paired species generate a level-$(k+1)$ Chern-Simons term for $c_\uparrow$. The resulting effective Lagrangian is

$$L_{\text{eff}} = -\frac{k+2}{4\pi}c_\uparrow dc_\uparrow - \frac{k}{2\pi}Adc_\uparrow + \frac{k+1}{4\pi}c_\uparrow dc_\uparrow + L_{\text{FL}}[f,a] + \frac{1}{4\pi}(a-c_\uparrow)d(a-c_\uparrow) + 2\text{CS}_g$$
$$= L_{\text{FL}}[f,a] + \frac{1}{4\pi}ada - \frac{1}{2\pi}adc_\uparrow - \frac{k}{2\pi}Adc_\uparrow + 2\text{CS}_g. \tag{44}$$

Integrating out $c_\uparrow$ sets $a = -kA$ and further simplifies the Lagrangian to

$$L_{\text{odd}-k} = L_{\text{FL}}[f,-kA] + \frac{k^2}{4\pi}AdA + 2\text{CS}_g. \tag{45}$$

With no emergent gauge field in sight, the Lagrangian in (45) describes an ordinary Fermi liquid made up of charge-$k$ fermions coexisting with a period-$(k+2)$ charge density wave. Note that the background integer quantum Hall response comes with an unusual coefficient which can be rewritten as

$$L_{\text{background}} = \frac{k^2}{4\pi}AdA + 2k^2\text{CS}_g - (2k^2-2)\text{CS}_g. \tag{46}$$

When $k = 1$, this reduces to $\text{CS}[A,g]$, which is the topological response of a single filled Landau level. For other odd $k$ with $k > 1$, the first term can be interpreted as $k^2$ filled Landau levels. As for the second term, we note that $2k^2 - 2 = 2(k+1)(k-1)$ is always a multiple of 16. Therefore, the additional gravitational Chern-Simons term can be accommodated by stacking with $(k^2-1)/8$ copies of the bosonic $E_8$ state.

Our results in the "ferromagnetic" case are highly exotic but admit a simple interpretation from the perspective of anyon fusion. Returning to the "ferromagnetic" Lagrangian

$$L_{\text{ferro}} = -\frac{k(k+2)}{4\pi}c_\uparrow dc_\uparrow - \frac{k}{2\pi}Adc_\uparrow + \sum_{\alpha=1}^{k+2}L[\Phi_{\alpha,\uparrow},c_\uparrow], \tag{47}$$

we see that local operators charged under the external gauge field $A$ are monopole operators of $c_\uparrow$, with fluxes quantized in integer multiples of $2\pi$. The mutual Chern-Simons coupling between $A$ and $c_\uparrow$ assigns a physical charge $k$ to the minimal bare monopole $\mathcal{M}_c$ which inserts $-2\pi$ flux of $c_\uparrow$. Due to the self Chern-Simons term for $c_\uparrow$, the bare monopole $\mathcal{M}_c$ carries charge $k(k+2)$ under $c_\uparrow$ and fails to be gauge-invariant. Dressing the bare monopole with $k(k+2)$ matter field operators, we obtain the basic gauge-invariant monopole, schematically written as $\tilde{\mathcal{M}}_c = \mathcal{M}_c\Phi^{k(k+2)}$. In the algebraic language, this charge-$k$ local excitation can be generated from the fusion product $a_0^{k(k+2)}$ where $a_0$ is the basic Abelian anyon sourced by $\Phi_\alpha$ in the "ferromagnetic" Lagrangian.

In a lattice electronic system, local excitations with even/odd charge are always bosons/fermions. When $k$ is even, $\tilde{\mathcal{M}}_c$ creates a local charge-$k$ boson. All finite-energy local excitations in this system therefore carry charges in integer multiples of $k$ and charged fermion excitations are projected out of the IR spectrum. From this perspective, a charge-$k$ superconductor naturally emerges. When $k$ is odd, $\tilde{\mathcal{M}}_c$ instead creates a local charge-$k$ fermion. Within the mean-field limit, we have the approximate relation $\Delta_{\text{Cooper}} \approx 2\Delta_{\text{electron}}$. Since like charges repel, gauge fluctuations tend to increase $\Delta_{\text{Cooper}}$ relative to the mean-field estimate (when the charges are a finite distance apart), thereby disfavoring a superconducting state for odd $k$. This heuristic reasoning based on the spectrum of monopole operators indeed reproduces the field-theoretic results in (41) and (45).

At sufficiently large $k$, the charge-$k$ superconductors and Fermi liquids that we have constructed are likely unstable. In the ferromagnetic Lagrangian (39), each boson species $\Phi_\alpha$ only feels a small magnetic field that is a fraction $1/k$ of its density. Therefore, a bosonic superfluid state for $\Phi_\alpha$ likely wins over the bosonic IQH/CFL state considered earlier. This superfluid has several unusual properties. First, the condensation of $\Phi_\alpha$ Higgses the gauge field $c_\uparrow$ completely, thereby destroying the parent topological order. Since $\Phi_\alpha$ transforms projectively under lattice translations with a projective phase $e^{2\pi i/(k+2)}$, the condensation of $\Phi_\alpha$ also leads to the formation of a CDW with period $(k+2)$. Moreover, since the $\Phi_\alpha$ is at Landau level filling $k$, the formation of the superfluid is accompanied by a vortex lattice with period $k$ associated with each $\Phi_\alpha$. Therefore, instead of a charge-$k$ superconductor or a charge-$k$ Fermi liquid, the likely fate of the doped system at large $k$ is a "crystal of crystal" with two distinct periodicities $k, k+2$ and no residual topological order.

## 4.2 Doping other minimally charged anyons

When $k > 2$, the $\mathrm{RR}_k$ state has multiple anyons that carry the same minimal charge $e/(k+2)$. In the $U(2)$ CSGL theory, these anyons correspond to $U(2)$ irreducible representations $(j, 1)$ with $0 < j \leq k/2$ and $j \in \mathbb{Z}+1/2$. For each allowed value of $j$, the associated CSGL Lagrangian takes the form

$$L = L_{\mathrm{RR}_k}[c] + \sum_{\alpha=1}^{k+2} L_{(j,1)}[\Phi_\alpha, c], \qquad (48)$$

where each bosonic matter field $\Phi_\alpha$ transforms in the $(j, 1)$ representation of $U(2)$. At nonzero doping, one can then consider different mean-field solutions and determine the itinerant phase that they describe. For $j \neq 1/2$, this task is difficult in general, as each $\Phi_\alpha$ is a $(2j + 1)$ dimensional vector and many possible mean-field states are available. In this section, we will not pursue an exhaustive classification of the resulting phases for all $k, j$. Instead, we will describe the examples with $k$ odd, and dope the $(j = k/2, n = 1)$ quasiparticle which is an Abelian anyon. We will argue that the resulting itinerant phases necessarily have coexistent non-Abelian topological order. For instance, for $k = 3$, doping this anyon leads to a period-5 CDW metal with ordinary electronic quasiparticles coexisting with a gapped electrically neutral Fibonacci topological order.

These results follow from a factorization property of $\mathrm{RR}_k$ for odd $k$, where $\mathrm{RR}_k = \mathcal{F}_k \times \mathcal{V}^{k+2,k}$ [48, 49]. In this formula, $\mathcal{F}_k$ denotes the neutral non-Abelian topological order generated by integer-$j$ Wilson lines of $SU(2)_k$ and $\mathcal{V}^{k+2,k}$ denotes the charged Abelian topological order generated by the vison in $\mathrm{RR}_k$ with charge $k/(k+2)$ and spin $k/2(k+2)$. Among all the anyons of $\mathrm{RR}_k$ that carry the minimal charge $1/(k+2)$, the unique Abelian one is $(k/2, 1)$, which maps to the unique minimally charged anyon $x$ in $\mathcal{V}^{k+2,k}$. If doping $x$ in $\mathcal{V}^{k+2,k}$ gives a phase A, then doping $(k/2, 1)$ in $\mathrm{RR}_k$ simply gives A*, where the star indicates the coexistence of A with a neutral $\mathcal{F}_k$ non-Abelian topological order.

Let us now specialize to $k = 3$, where $\mathrm{RR}_3 = \mathcal{F}_3 \times \mathcal{V}^{5,3}$. Now the charged Abelian sector $\mathcal{V}^{5,3}$ is equivalent to the Jain state at filling $\nu = 3/5$. The non-Abelian sector is precisely the Fibonacci theory with a single non-Abelian anyon $\tau$ satisfying the fusion rule $\tau \times \tau = 1 + \tau$. If we label the minimally charged anyon in the $\nu = 3/5$ Jain state as $x$, then the anyons $(1/2, 1), (3/2, 1)$ in the $U(2)$ formulation map to $\tau x, x$ in the factorized formulation respectively.

With this factorization, we see that doping $(3/2, 1)$ in the $\mathrm{RR}_3$ state is equivalent to doping the charge $1/5$ anyon $x$ in the $\nu = 3/5$ Jain state while leaving the Fibonacci sector invariant. As shown in Appendix B.2 of Ref. [8], the doped Jain state realizes a Fermi liquid metal with period-5 charge order. From here, we immediately conclude that doping $(3/2, 1)$ in the $\mathrm{RR}_3$

state induces a CDW FL* state, in which a period-5 charge-ordered Fermi liquid coexists with a neutral Fibonacci topological order.

For more general choices of minimally charged anyons and/or even $k$, the factorization property does not apply and we must work with the $U(2)$ CSGL theory and study matter fields in higher spin representations of $U(2)$. This is an interesting class of theoretical problems that we will defer to future work.

# 5 Charge-flux unbinding: Physical picture for doped FQAH insulators

While the field-theoretic formalism in Section 3 and Section 4 enables a clean derivation of various itinerant phases in the doped FQAH insulator, it provides limited intuition for their physical origin. Consider, for example, any of the superconducting states that arise from doping an Abelian/non-Abelian parent insulator. A naive picture for the origin of superconductivity is that fractionally charged anyons bind into a local boson which subsequently condenses.[8] However, this picture cannot be correct because condensing the local boson leaves the parent topological order intact and results in an SC* state. In order to obtain an ordinary SC state with no residual topological order, we must condense a fractionally charged boson that destroys the entire parent topological order. Unfortunately, such a quasiparticle simply does not exist in a general FQAH insulator. Even in the simplest fermionic Laughlin state at $\nu = 1/3$, every fractionally charged anyon also carries fractional statistics. Although a single anyon has a well-defined dispersion, the collective motion of a finite density of anyons is frustrated by the nontrivial braiding phases and the anyons cannot condense directly. With no fractionally charged bosons in sight, how is it possible to interpret the SC states in terms of condensation?

## 5.1 General framework for doped Abelian states

A partial resolution of this puzzle emerges out of a parton construction of doped FQAH insulators that is conceptually different from the CSGL approach in earlier sections of this work. Let us first illustrate this construction for the simplest superconductor that arises from doping a $U(1)_2$ state realized by microscopic charge-1 bosons at lattice filling $\nu = 1/2$. When semions are the cheapest charged excitations, the CSGL Lagrangian for the doped state takes the form

$$L = -\frac{2}{4\pi}\alpha d\alpha + \frac{1}{2\pi}\alpha dA + L[\phi, \alpha], \tag{49}$$

where $\alpha$ is an emergent $U(1)$ gauge field and $A$ is the external electromagnetic gauge field. We drop the species index in $\phi$, as translation symmetry fractionalization does not play a role in this discussion. At nonzero doping, the equations of motion for $\alpha$ imply that $\phi$ is at filling 2 with respect to the emergent magnetic field generated by $\alpha$. Therefore, $\phi$ can go into the bosonic IQH state and the remaining Lagrangian describes a trivial charge-1 superfluid.

We can give an alternative construction of the same state by writing the scalar field $\phi$ as $\phi = \Phi\tilde{\phi}$. This parton construction requires the introduction of a new $U(1)$ gauge field $\beta$ to project out unphysical states. The resulting Lagrangian is then

$$L = -\frac{2}{4\pi}\alpha d\alpha + \frac{1}{2\pi}\alpha dA + L[\Phi, \alpha - \beta] + L[\tilde{\phi}, \beta]. \tag{50}$$

---

[8]Throughout this section, we refer to a quasiparticle as a "boson" if it has bosonic self-statistics. Such a boson could still have nontrivial braiding with other anyons in the topological order. This is distinguished from a "local boson", which refers to a trivial boson that is topologically identified with the vacuum.

Now imagine introducing excess bosons with lattice filling $\delta$. The mutual CS term between $\alpha$ and $A$ enforces the constraint that

$$\frac{1}{2\pi}\nabla \times \boldsymbol{\alpha} = \delta\,, \quad \rho_\phi = \rho_\Phi = \rho_{\tilde{\phi}} = 2\delta\,. \tag{51}$$

In the doped theory, we can choose a mean-field state in which the flux of $\beta$ adjusts itself to match the flux of $\alpha$. This choice liberates the kinetic energy of $\Phi$, as it now sees zero magnetic field. The boson $\tilde{\phi}$ is at Landau level filling 2 and naturally forms the bosonic IQH state. Therefore, the Lagrangian $L[\tilde{\phi},\beta]$ describes a neutral $U(1)_{-2}$ topological order that is stacked with the parent charged $U(1)_2$ topological order. The $\Phi$ field sources an anyonic exciton which is a semion-antisemion pair in $U(1)_2 \times U(1)_{-2}$. The condensation of $\Phi$ then kills the topological order and gives a trivial charge-1 superfluid.

The physical idea demonstrated by the parton construction above is that doping can give rise to anyons that eventually "unbind" into fractionally charged bosons and neutral anyons. The subsequent condensation of these fractionally charged bosons then has two effects. First, due to the charge of the bosons, we get a superconductor. Second, the condensation of the bosons changes the topological order, reducing the total quantum dimension. This picture can be interpreted as a sequence of two topological manipulations: first, we "stack" with a charge neutral topological order to get the resulting theory after unbinding ($U(1)_2 \times U(1)_{-2}$ if we start with $U(1)_2$), and then we condense a set of charged bosonic anyons (the bound state of the semion and the anti-semion), resulting in a superconductor (in this case, with trivial topological order because everything else gets confined). Note that the doped system realizes both of these steps–the charge-flux unbinding and condensation–simultaneously. There is no separation in parameter space or even in length/energy scales between these processes.

It is straightforward to generalize the above construction to an arbitrary doped Abelian state enriched by a global $U(1)$ symmetry. The parent topological order can be described by a $K$-matrix theory

$$\mathcal{L}_p = -\frac{1}{4\pi}\sum_{I,J} K_{IJ} b_I d b_J + \frac{1}{2\pi}A\sum_I q_I d b_I\,, \tag{52}$$

where $b_I$ are $U(1)$ gauge fields and $q_I$ is an integer charge vector. Anyons are labeled by an integer $l$-vector. Their self-statistics is $\theta = \pi l^T K^{-1} l$ and their electric charge is $q^T K^{-1} l$. We consider doping this state by introducing a finite density of anyons with some particular $l$. The doped state can be described by introducing a bosonic matter field $\phi$ with a Lagrangian

$$\mathcal{L} = \mathcal{L}_p + \mathcal{L}[\phi, \sum_I l_I b_I]\,. \tag{53}$$

The density of $\phi$ is determined by the density of doped charges (and the fractional charge of the anyon nucleated by $\phi$). The equation of motion of $b_{I0}$ determines the mean magnetic field seen by $\phi$

$$\frac{1}{2\pi}\sum_J K_{IJ}\langle\nabla \times b_J\rangle = l_I J_0^\phi \quad \rightarrow \quad B_\phi = l_I\langle\nabla \times b_I\rangle = l^T K^{-1} l J_0^\phi\,. \tag{54}$$

If the self statistics of the anyon nucleated by $\phi$ is $l^T k^{-1} l = \pi n/m$, then it follows that $\phi$ is at a filling fraction $\nu_\phi = m/n$.

Following the construction for $U(1)_2$, we now introduce a parton construction $\phi = \Phi\tilde{\phi}$, and write a Lagrangian with a new dynamical gauge field $\beta$

$$\mathcal{L} = \mathcal{L}_p + \mathcal{L}[\Phi, \sum_I l_I b_I - \beta] + \mathcal{L}[\tilde{\phi},\beta]\,. \tag{55}$$

As usual in this parton construction, the densities of these partons will be equal ($J_0^\phi = J_0^\Phi = J_0^{\tilde\phi}$). Now we wish to choose the mean value of $\beta$ such that the total mean magnetic flux seen by $\Phi$ vanishes:

$$\langle \nabla \times \beta \rangle = B_\phi . \tag{56}$$

The $\tilde\phi$ will see an effective magnetic field and will be at a Landau level filling $\nu_{\tilde\phi} = m/n$. To develop a "stack and condense" picture, we perform a particle-vortex duality on $\tilde\phi$ to a boson field $\widehat\phi$ to write the Lagrangian

$$\mathcal{L} = \mathcal{L}_p + \mathcal{L}[\Phi, \sum_I l_I b_I - \beta] + \frac{1}{2\pi}\widehat{b}d\beta + \mathcal{L}[\widehat\phi, \widehat{b}] . \tag{57}$$

The first term describes the parent topological order $T_p$. The last two terms describe some topological order $T_s$. Since $\tilde\phi$ is at filling $m/n$, the particle-vortex dual $\widehat\phi$ is at filling $\nu_{\widehat\phi} = -n/m$. Thus, $T_s$ can be regarded as a topologically ordered state of bosons at a Landau level filling $\nu_{\widehat\phi} = -n/m$. $\Phi$ nucleates an anyon that is a composite of the anyon $l_I$ from $T_p$ with self-statistics $\pi n/m$ and the vison $v_s$ in $T_s$ with self-statistics $-\pi n/m$. In the doped theory, $\Phi$ is at finite density and sees zero magnetic field. Thus a Higgs transition induced by condensing $\Phi$ will lead to a superconductor that spontaneously breaks the global $U(1)$ symmetry. Depending on the details of $T_p$ and $T_s$ this superconductor may have some coexisting residual topological order.

Let us now discuss the realization of microscopic lattice translation symmetry. At a lattice filling $\nu = p/q$, the (IR image of) the lattice $T_x$ and $T_y$ acts projectively on an anyon $a$:

$$T_x T_y T_x^{-1} T_y^{-1}[a] = e^{2\pi i B(a,a_b)}[a] , \tag{58}$$

where $a_b$ is the background anyon with global $U(1)$ charge $p/q$, and $B(a, a_b)$ is the braiding phase between $a$ and $a_b$. For the following discussion, we restrict to parent FQAH phases where $a_b$ is just the vison. Then $B(a, a_b) = Q(a) \mod 1$ where $Q(a)$ is the global $U(1)$ charge of $a$. Rather than developing the most general theory, let us focus on the case where $q$ is odd, and the full topological order is just generated by the vison. This theory has $q$ anyons $(1, v, v^2, \ldots, v^{q-1})$ and we denote it $\mathcal{V}^{q,p}$. Any anyon $v^r$ has charge $\frac{rp}{q}$ and spin $h_r = \frac{pr^2}{2q}$. The general projective action of translations shows that there is a non-trivial action of a $\mathbb{Z}_q \times \mathbb{Z}_q$ quotient of the full $\mathbb{Z} \times \mathbb{Z}$ UV translation symmetry (as expected from filling constraints). Thus we can regard the FQAH state as a topological ordered state enriched with both global $U(1)$ and $\mathbb{Z}_q \times \mathbb{Z}_q$ symmetries.

In the IR CSGL theory, this $\mathbb{Z}_q \times \mathbb{Z}_q$ symmetry acts as an internal symmetry. Now let us consider doping the theory with an anyon $a = v^r$. For simplicity we further restrict attention to $r$ such that $gcd(r, q) = 1$. Then $a$ can equally well be regarded as a generator of the $\mathcal{V}^{q,p}$ theory.

Let us go through the route of describing this by a boson $\phi$ coupled to U(1) gauge fields, and then doing partons $\phi = \Phi\tilde\phi$ as described above. Finally let us do particle-vortex duality on $\tilde\phi$ to get vortices $\widehat\phi$ at a Landau level filling $-\frac{pr^2}{q}$. The corresponding bosonic fractional quantum Hall state defines the stacked theory $T_s$ and we want to condense the bound state of its vison $v_s$ and the anyon $a$ to get a superconductor. We now argue that (at least with the restrictions discussed above) we can always symmetry enrich $T_s$ so that the projective phase of $v_s$ under $\mathbb{Z}_q \times \mathbb{Z}_q$ is exactly the opposite of that of $a$ in the parent TO.

To see this, first note that $T_s$ has a subset of Abelian anyons generated by $v_s$. As $h(v_s) = -\frac{pr^2}{2q}$, there are (at least) $q$ such anyons $(1, v_s, v_s^2, \ldots, v_s^{q-1})$. Clearly as we assume $a$ is a generator of $\mathcal{V}^{q,p}$, it follows that the anyons generated by $v_s$ contains $\bar{\mathcal{V}}^{q,p}$ where the

overline denotes time-reversal conjugation. This means it is always possible to give $v_s$ the conjugate symmetry enrichment under $\mathbb{Z}_q \times \mathbb{Z}_q$ by assigning the projective phase to be $B(v_s, \tilde{a_b})$ where $\tilde{a_b}$ is the anyon corresponding to the conjugate of the background anyon of the parent FQAH state.

Thus it is always possible (with the restrictions above) for the stacked theory $T_s$ to be symmetry enriched such that the condensation of the boson $av_s$ preserves $\mathbb{Z}_q \times \mathbb{Z}_q$. Then we get a translation symmetry preserving SC. This SC may not be charge-2, and may be either an SC or an SC*, depend on other choices but at least under fairly general conditions, the lattice translation can be preserved.

We can illustrate this general framework through several superconductors in the doped FQAH regime. Any Abelian topological order can be written as a trivial fermionic theory with excitations $\{1, f\}$ stacked with a bosonic topological order. For example, the Abelian $v = \frac{1}{3}$ state can be written as a trivial fermionic theory with $c_- = 3$ stacked with the bosonic theory $\mathcal{C} = U(1)_{-6} \times U(1)_{-2}/\mathbb{Z}_2$. $\mathcal{C}$ has chiral central charge $c_- = -2$, so that the total chiral central charge is $c_- = 1$. It has three anyon types $a, a^2, 1$ with topological spins $2/3, 2/3, 0$ respectively, obeying a $\mathbb{Z}_3$ fusion rule.[9] The vison is given by the bound state of the fermion and the generator $a$ of $\mathcal{C}$, which has spin $1/6$, and $a$ by itself has charge $2/3$. Doping $a$ according to the above story can then lead to charge-flux unbinding, resulting in $\tilde{\phi}$ forming a charge neutral $\bar{\mathcal{C}}$ layer. $a\bar{b}$ with $a \in \mathcal{C}$ and $\bar{b} \in \bar{\mathcal{C}}$ is a charged boson, whose condensation results in a trivial superconductor with $c_- = 1 + 2 = 3$. The same stack and condense procedure applies to the doped $v = 2/3$ state in Ref. [8]. In this case, we start with the particle hole conjugate of the $v = 1/3$ state, i.e. $U(1)_1 \times U(1)_{-3}$. The resulting superconductor has chiral central charge given by particle hole conjugation as $1 - 3 = -2$. This resulting chiral central charge can also be derived from stacking with $\mathcal{C}$ rather than $\bar{\mathcal{C}}$ because we start with $U(1)_{-3}$ rather than $U(1)_3$. This decreases the $c_-$ by two from its original value $c_- = 0$ in the $v = 2/3$ state, which agrees with the field-theoretic calculations in Ref. [8].

Finally, we remark that the parton construction in (57) in fact applies even when the parent theory is non-Abelian, so long as the doped anyon sourced by $\phi$ lives in an Abelian subsector. As a simple example, we can consider doping the Abelian vison $a^2$ in the non-Abelian Moore-Read (Pfaffian) state. The effect of doping is to unbind the vison into a bosonic charge $a^2\bar{s}$ and a flux $s$ with semionic statistics, obtained by stacking with an anti-semion theory $\{1, \bar{s}\}$. Condensing the fractionally charged bound state $a^2\bar{s}$ leads to a trivial superconductor with chiral central charge $c_- = \frac{3}{2} - 1 = \frac{1}{2}$. The condensation undoes the flux attachment because $\psi a^4$ gets identified with $\psi$, resulting in a $p + ip$ topological superconductor, in agreement with the field-theoretic calculations in Section 3.3.

## 5.2 Comments on generalizations to doped non-Abelian states

For doped Abelian states, the intuition of charge-flux unbinding led us to a "stack and condense" algebraic procedure. This algebraic procedure naturally generalizes beyond the Abelian setting. Given any parent non-Abelian fermionic topological order $T_p$ in which the anyon $a_0 \in T_p$ has the smallest energy gap, doping-induced superconductors can be described by first stacking with a neutral bosonic topological order $T_s$ and then condensing a charged boson $a_0 b \in T_p \times T_s$ (more precisely, a condensable algebra $\mathcal{B}$ that contains $a_0 b$ [50–53]). The resulting state is a superconductor coexisting with a reduced topological order $[T_p \times T_s]/\mathcal{B}$ (which can be trivial). When $T_p$ is the Moore-Read state, we can obtain a superconductor

---

[9]To see this, note that $U(1)_{-6}$ has 6 anyons with spins $-n^2/12$ and $U(1)_{-2}$ has two anyons with spins $-m^2/4$. The $/\mathbb{Z}_2$ indicates condensation of $(n = 3, m = 1)$, which is a boson: $-9/12 - 1/4 = 0 \mod 1$. The remaining anyons are $(n = 1, m = 1), (n = 2, m = 0), (n = 3, m = 1)$ and the last anyon type is identified with the vacuum because it is condensed. These have spins $2/3, 2/3, 0$ respectively. The generator of $U(1)_3$ is then given by $(n = 1, m = 1)$ together with the fermion; it has spin $4/6 + 1/2 = 1/6 \mod 1$.

with arbitrary chiral central charge by stacking with a $T_s$ which is (the time-reversal conjugate of) one of the 16 minimal modular extensions of the Moore-Read state (see Appendix E). For example, the $c_- = -1/2$ topological superconductor in Section 3.1 can be recovered with $T_s = \left[ \overline{\text{Ising}} \times \overline{\text{Ising}} \times U(1)_{-8} \right] / \mathbb{Z}_2$.

Mimicking the Abelian framework, we can use a parton construction to describe an arbitrary doped non-Abelian state and attempt to draw a connection between the parton description and the "stack and condense" procedure. Consider a parent topological order $T_p$ described by some non-Abelian gauge theory $L_{\text{parent}}[\alpha, A]$ where $\alpha$ is a $G$ gauge field and $A$ is a background $U(1)$ gauge field. If we dope a non-Abelian anyon $a_0$ in the irreducible representation $R$ of $G$, then the doped CSGL theory takes the form

$$L = L_{\text{parent}}[\alpha, A] + L_R[\phi, \alpha], \tag{59}$$

where $\phi$ is a scalar field that transforms in the irreducible representation $R$ of $G$. In general, we can do a parton construction $\phi_i = \Phi_{ij} \tilde{\phi}_j$ where $\tilde{\phi}_j$ transforms in the $R'$ irreducible representation of some other gauge group $G'$. This means we need to introduce a $G'$ gauge field $\beta$ to project out unphysical states. The effective Lagrangian can then be written as

$$L = L_{\text{parent}}[\alpha, A] + L_{R \otimes \overline{R'}}[\Phi, \alpha \otimes I - I \otimes \beta] + L_{R'}[\tilde{\phi}, \beta]. \tag{60}$$

In the doped state, we can arrange the fluxes of $\alpha$ and $\beta$ such that $\Phi$ prefers to condense. With every choice of flux configuration, we can then consider different allowed gapped states for $\tilde{\phi}$. For each gapped state, $L_{R'}[\tilde{\phi}, \beta]$ defines a neutral bosonic topological order $T_s$. In this picture, the doped anyon sourced by $\phi_i$ indeed dissociates into a fractionally charged boson $\Phi_{ij}$ and a neutral anyon sourced by $\tilde{\phi}_j$. When the anyon sourced by $\Phi$ in $T_p \times T_s$ is part of a condensable algebra $\mathcal{B}$, the condensation of $\Phi$ realizes a superconductor with residual topological order $T_p \times T_s / \mathcal{B}$.

Unfortunately, the correspondence between this non-Abelian parton construction and the "stack and condense" procedure is much less transparent than the Abelian case. In the Abelian setting, the statement that $\Phi$ sees zero flux on average is always equivalent to the statement that $\Phi$ sources a condensable bosonic anyon in the theory $T_p \times T_s$. This equivalence fails to hold in the non-Abelian setting because there is no direct connection between the dynamical conditions for the field-theoretic condensation of $\Phi$ and the algebraic condensability of the anyon sourced by $\Phi$.

Even in situations where it is possible to choose $T_s$ such that $\Phi$ sources a condensable boson in $T_p \times T_s$, the parton construction does not provide a recipe for identifying a preferred $T_s$. In the Abelian setting, we showed that for any doped anyon $a_0 \in T_p$ with fractional statistics $\pi n/m$, $L_{R'}[\tilde{\phi}, \beta]$ describes a topological order $T_s$ for the vortex field $\widehat{\phi}$ at a fictitious $U(1)$ filling $-n/m$. This vortex perspective allows us to classify the possible choices of $T_s$ and identify conditions under which any $T_s$ could be stabilized energetically. However, when $\beta$ is a non-Abelian gauge field, there is no particle-vortex duality in general and $T_s$ cannot be interpreted as the topological order formed by vortices of $\tilde{\phi}$ at a specific filling factor of the non-Abelian symmetry $G'$. This fact obscures the dynamical origin of $T_s$ and further compromises the utility of the parton construction.

In summary, the intuition of charge-flux unbinding does not generalize to the non-Abelian setting in a straightforward way. The field-theoretic calculations in Section 3 and Section 4 therefore do not admit a simple "stack and condense" interpretation with a corresponding parton description. Providing an alternative physical picture for doped non-Abelian states remains an important open problem that we hope to revisit in the future.

# 6 Discussion

The problem of a doped FQAH insulator features a rich interplay between interactions and fractional statistics in the gapless regime. In this work, generalizing the constructions of Ref. [8] for Abelian Jain states, we studied a sequence of doped non-Abelian Read-Rezayi states $RR_k$, which include the Moore-Read state ($k = 2$) as a special case. An important conceptual feature of the non-Abelian cases is that doping induces an energetic splitting between anyon fusion channels that are degenerate in the undoped parent topological phase. Incorporating this energy splitting into our field-theoretic analysis, we found an intriguing even/odd effect in the Read-Rezayi index $k$. When $k$ is even, a superconductor always emerges at nonzero doping, with its flux quantization and topological properties determined by the energy splitting. When $k$ is odd, depending on the choice of preferred fusion channel, doping can stabilize an ordinary BCS superconductor or a charge-$k$ Fermi liquid with concomitant period-$(k + 2)$ charge density wave. This charge-$k$ Fermi liquid arises out of pairing instabilities in an intermediate-temperature non-Fermi liquid state with no well-defined quasiparticles.

The various superconducting and metallic phases realized by the doped non-Abelian anyon fluid have sharp experimental signatures. For all $k$, the paramagnetic solution always gives a topological charge-2e superconductor with chiral central charge $c_- = -1/2$. On a sample with boundary, this chiral central charge implies a single counter-propagating Majorana edge mode which contributes a quantized thermal Hall conductance $\kappa_{xy} = -\pi^2 k_B^2 T/6$. In the presence of quenched impurities, a generalization of the analysis in [54] shows that, across the FQAH-SC phase transition, localized non-Abelions with charge $e/(k + 2)$ in the $RR_k$ plateau evolve into vortices of the topological superconductor with flux $h/2e$ or $-(k + 1)h/2e$ (see Appendix F). The superconducting state near the transition thus realizes an anomalous vortex glass phase with randomly pinned $h/2e$ and $-(k+1)h/2e$ vortices, such that the total vorticity vanishes. The dependence of the preferred vorticities on $k$ allows us to distinguish the topological superconductors that arise from doping different $RR_k$ states, despite the fact that they all share the same chiral central charge in the clean limit. Moreover, due to the half-integer chiral central charge, each vortex traps a Majorana zero mode and the interactions between these zero modes could lead to a thermal metal with large longitudinal thermal conductance $\kappa_{xx}$ [55–58]. A direct observation of this vortex glass phase would provide strong evidence for the anyon-driven mechanism for superconductivity.

In contrast to the paramagnetic solutions, the ferromagnetic solutions generally do not have quantized Hall response or protected edge states. For even $k$, the ferromagnetic solution gives a charge-4e non-topological superconductor with flux quantization in units of $h/(ke)$. This unusual flux quantization can be detected via standard measurements such as the fractional Josephson effect. Following the analysis in [54], the elementary charge $e/(k+2)$ anyon evolves into vortices of the superconductor with flux $h/ke$ and $-(k+1)h/ke$ (see Appendix F). The superconductor near the FQAH-SC transition is therefore also an anomalous vortex glass, although its vortices do not trap Majorana zero modes. For odd $k$, the ferromagnetic solution gives a charge-$k$ Fermi liquid with a period-$(k+2)$ CDW order. In tunneling spectroscopy, this phase exhibits a nonzero energy gap to local excitations with charge $Q < k$, and a well-defined Fermi surface made up of charge-$k$ fermionic quasiparticles. Moreover, the charge-ordering gives rise to Bragg peaks at wavevectors with magnitude $2\pi/(k+2)$, although the precise orientation of the ordering wavevector depends on non-universal parameters in the microscopic Hamiltonian.

Our analysis of the doped non-Abelian states raises a series of open questions. On the field-theoretic side, strongly coupled non-Abelian CSGL theories of the type introduced in Section 2 are difficult to solve exactly. In this work, we made progress by starting with the simplest translation-preserving mean-field solutions consistent with LSM constraints (motivated by the

splitting of fusion channels) and exactly analyzing fluctuations around them.[10] It would be interesting to search for a "weak-coupling" limit in which this approach can be justified rigorously. Following the existing literature on Chern-Simons matter theories, it may be fruitful to embed the $U(2)_{k,-2(k+2)}$ theory in a larger family of $U(N)_{k_1,k_2}$ theories and consider a solvable deformation in which the rank $N$ and/or the levels $k_1, k_2$ are taken to infinity [59–62]. Although these deformations are not directly relevant to experimentally realized FQAH insulators, they would provide a solid proof-of-principle for the novel itinerant phases in our work and may offer theoretical insights that generalize away from the large rank/level limit.

Orthogonal to the pursuit of field-theoretic refinements is the search for a cleaner physical intuition for the emergence of metallic and superconducting phases in the doped FQAH regime. On the superconducting side, we made a first attempt towards elucidating their physical origin in Section 5 from the perspective of charge-flux unbinding. The basic idea is to regard a general anyon as a charge-flux bound state. The effect of doping is to dissociate the bound state into a fractionally charged boson and a neutral flux that carries the fractional statistics. The subsequent condensation of the fractionally charged boson confines all anyons that braid with it and drives the formation of superconductivity. While this perspective provides an appealing intuitive picture and an efficient tool for doping complicated topological orders, it does not answer the dynamical question of why the dissociation occurs in the first place. Without a solution to this dynamical question, the framework of charge-flux unbinding remains incomplete.

The metallic phases in the doped FQAH regime are even more counterintuitive. For odd $k$, doping the $RR_k$ Read-Rezayi state produces a highly exotic charge-$k$ Fermi liquid with period-$(k+2)$ charge order. From the pioneering work by Read and Rezayi, we know that the $RR_k$ topological order can be realized in the lowest Landau level with a contact $(k+1)$-body repulsive interaction that penalizes the clustering of more than $k+1$ electrons [28]. To minimize the energy cost from this interaction, electrons avoid forming clusters with size greater than $k$. However, local fermions with charge less than or equal to $k$ are not penalized and it is not clear why a charge-$n$ Fermi liquid with $n < k$ does not appear in our analysis. This tension suggests the existence of additional correlation effects in the $RR_k$ states that favor the $k$-clustering of electrons, which calls for further investigation.

In parallel to analytical treatments, it would be tremendously fruitful to also explore doped non-Abelian states through large-scale numerical studies. Since the discovery of Abelian FQAH insulators in both twisted $MoTe_2$ and rhombohedral multilayer graphene, novel non-Abelian phases have been proposed in a variety of moire materials and established through extensive numerical simulations on approximate microscopic models [12–17]. The analytical results in this work provide a strong motivation to study the same microscopic models at nonzero doping. While exact diagonalization is limited to relatively small system sizes where it is only meaningful to dope in a small number of electrons, it may be possible to approach the thermodynamic limit through density-matrix renormalization group methods [63–65] and/or variational wavefunctions constructed out of modern neural networks [66–69]. In addition to predicting itinerant phases, these techniques may also access the energy spectra of individual anyons and the energy splitting of distinct anyon fusion channels. A numerical study of the connection between few-anyon energetics and the many-body phase in the anyonic fluid could serve as a direct test of the analytical heuristics outlined in Section 2.4 and Section 3.

Finally, the ultimate dream is to study itinerant anyonic quantum matter in real experimental systems. A remarkable feature of many moire materials is the possibility to tune the spatial range of electronic interactions through gate screening. When the screening length is sufficiently small, Wigner crystallization is disfavored and gapless itinerant phases could emerge

---

[10]For example, when fermions/bosons are at an integer/even-integer filling, we put them in the fermionic/bosonic integer quantum Hall state rather than more complicated states with topological order.

in some range of doping. We hope that theoretical progress on the open questions above will motivate a more detailed experimental characterization of the doped regime in non-Abelian FQAH insulators, should such phases be discovered in the future.

## Acknowledgments

We thank Ahmed Abouelkomsan, Meng Cheng, Tonghang Han, Patrick Ledwith, Aidan Reddy, Leyna Shackleton, Donna Sheng, Ashvin Vishwanath, Chong Wang, and Yunchao Zhang for discussions.

**Funding information**  Part of this work was completed during the KITP program "Generalized Symmetries in Quantum Field Theory: High Energy Physics, Condensed Matter, and Quantum Gravity", which is supported in part by grant NSF PHY-2309135 to the KITP. ZDS and TS are supported by NSF grant DMR-2206305. C.Z. is supported by the Harvard Society of Fellows.

## A  Review of Chern-Simons descriptions for non-Abelian quantum Hall states

In this Appendix, we review aspects of non-Abelian Chern-Simons theory that play an important role in the field-theoretic constructions of the main text. Our focus is on Chern-Simons theories involving a single $U(2)$ gauge field, which provide a unifying description of a large class of non-Abelian quantum Hall states. We will begin with an abstract introduction to $U(2)$ Chern-Simons theories in Appendix A.1, and apply the formalism to a few concrete examples in Appendix A.2. As a bonus, we will also provide in Appendix A.3 a derivation of the Moore-Read state from $p + ip$ pairing of composite fermions, using the formalism of $U(2)$ Chern-Simons theory.

### A.1  General features of $U(2)$ non-Abelian Chern-Simons theories

The most general Chern-Simons theory of interest to us is $U(2)_{k_1,k_2} = \left[ SU(2)_{k_1} \times U(1)_{k_2} \right] / \mathbb{Z}_2$. The Lagrangian for the product theory $SU(2)_{k_1} \times U(1)_{k_2}$ is easy to write down in terms of an $SU(2)$ gauge field $\beta$ and a $U(1)$ gauge field $\alpha$:

$$L_{SU(2)_{k_1} \times U(1)_{k_2}} = -\frac{k_1}{4\pi} \text{Tr} \left( \beta d\beta + \frac{2}{3} \beta^3 \right) - \frac{k_2}{4\pi} \alpha d\alpha. \tag{A.1}$$

In the $\beta_0 = \alpha_0 = 0$ gauge, the equations of motion impose the constraint that the field strength tensors associated with $\beta$ and $\alpha$ vanish identically. Therefore, the dynamical gauge-invariant operators in the theory are Wilson lines in various irreducible representations of $SU(2) \times U(1)$. These Wilson lines can be interpreted as world-lines of anyons in the underlying non-Abelian topological order, in the limit where the anyon masses are taken to infinity. We will refer to the anyon associated with the spin-$j$ representation of $SU(2)$ and charge-$n$ representation of $U(1)$ as $(j, n)$. By the standard Chern-Simons quantization rules, the non-redundant anyons of this theory satisfy $j \leq |k_1|/2$ and $n < |k_2|$ and have topological spin

$$h_{(j,n)}^{(k_1,k_2)} = \frac{j(j+1)}{k_1 + 2} + \frac{n^2}{2k_2}. \tag{A.2}$$

The fusion rules between distinct anyons match the fusion rules of $SU(2)$ and $U(1)$ representations, modulo a truncation on the maximum allowed $SU(2)$ and $U(1)$ representations:

$$(j_1, n_1) \times (j_2, n_2) = \sum_{j=|j_1-j_2|}^{\min(|k_1|/2, |k_1|-j_1-j_2)} (j, [n_1 + n_2 \bmod |k_2|]). \tag{A.3}$$

To go from the product theory $SU(2)_{k_1} \times U(1)_{k_2}$ to $U(2)_{k_1,k_2} = \left[SU(2)_{k_1} \times U(1)_{k_2}\right]/\mathbb{Z}_2$, we need to define a $\mathbb{Z}_2$ quotient. From the condensed matter perspective, this quotient is implemented by condensing an order-2 anyon in the parent product topological order, thereby confining all anyons that braid non-trivially with the condensing anyon. In order for the condensation to make sense, the topological spin of the condensing anyon must be an integer or a half-integer. When the condensing anyon has integer topological spin, we can directly condense it to get a bosonic quotient topological order. When the condensing anyon has half-integer topological spin, we need to stack with a trivial fermion theory $\{1, c\}$ and condense the bound state of the original anyon with the trivial fermion $c$. The resulting quotient theory is a fermionic topological order. Note that in this case the condensation of a self-boson (self-fermion) in a parent topological order $\mathcal{C}$ ($\mathcal{C} \times \{1, c\}$) yields the same result as gauging a $\mathbb{Z}_2$ one-form symmetry generated by an anyon with integer topological spin in $\mathcal{C}$ ($\mathcal{C} \times \{1, c\}$).

From the fusion rules, we see that the candidate order-2 anyon is labeled by $(j = k_1/2, k_2/2)$ with associated topological spin

$$h_{k_1/2, k_2/2}^{(k_1, k_2)} = \frac{k_1}{4} + \frac{k_2}{8} = \frac{2k_1 + k_2}{8}. \tag{A.4}$$

Consistent $U(2)_{k_1,k_2}$ theories therefore satisfy the additional quantization condition $2k_1 + k_2 \in 4\mathbb{Z}$.

At the level of Lagrangians, a convenient way to implement the $\mathbb{Z}_2$ quotient is to rewrite the Lagrangian in terms of a new $U(2)$ gauge field $a = \alpha I + \beta$ and sum over non-trivial principal $U(2)$-bundles rather than $SU(2) \times U(1)$ bundles in the path integral. Simple algebraic manipulations show that

$$L = -\frac{k_1}{4\pi} \operatorname{Tr}\left(ada + \frac{2}{3}a^3\right) + \frac{2k_1 - k_2}{4} \frac{1}{4\pi} \operatorname{Tr} a\, d \operatorname{Tr} a. \tag{A.5}$$

Since $2k_1 + k_2 \in 4\mathbb{Z}$ and $k_1, k_2 \in \mathbb{Z}$, the Chern-Simons levels in the two terms above both satisfy integer quantization. The anyons that survive the $\mathbb{Z}_2$ quotient satisfy $j + n/2 \in \mathbb{Z}$ and correspond to irreducible representations of $U(2)$. The Lagrangian in (A.5) is the most compact representation of the $U(2)_{k_1,k_2}$ Chern-Simons theory and will be used throughout the paper.

## A.2 Application to concrete examples

Let us illustrate the general framework of Appendix A.1 in several concrete examples. One of the simplest and most familiar non-Abelian topological order is the Ising topological order described by the $U(2)_{2,-4}$ Chern-Simons theory. Following the general recipe above, the anyons of $U(2)_{2,-4}$ are labeled by $(j, n)$ with $j \leq 1, n < 4$, $j + n/2 \in \mathbb{Z}$. Furthermore, since the Ising topological order is bosonic (i.e. $2k_1 + k_2 = 0 \mod 8$), the anyons $(j, n)$ and $(1 - j, (n - 2) \mod |k_1|)$ should be identified. Therefore, the only non-trivial anyons that survive are

$$1 = (0, 0), \quad \sigma = (1/2, 1), \quad \psi = (0, 2). \tag{A.6}$$

The fusion between these anyons can be worked out using (A.3) and agree with the familiar fusion rules for the Ising topological order that we stated in Section 2.1:

$$\psi \times \psi = 1, \quad \sigma \times \psi = \sigma, \quad \sigma \times \sigma = 1 + \psi. \tag{A.7}$$

The Ising topological order appears as a building block in the simplest non-Abelian fractional quantum Hall state: the Moore-Read (Pfaffian) state at filling $\nu = 1/2$. Following the notations of Ref. [23], we write the topological order of the Moore-Read state as

$$\text{Moore-Read} = \left[\text{Ising} \times U(1)_8\right]/\mathbb{Z}_2 = \left[U(2)_{2,-4} \times U(1)_8\right]/\mathbb{Z}_2. \tag{A.8}$$

Here, the anyons in the product theory are labeled by a $(j, n)$ representation of $U(2)$ and a charge-$m$ representation of $U(1)$. The quotient by $\mathbb{Z}_2$ is implemented by condensing a self-fermion $f \sim (j = 0, n = 2, m = 4)$. Though the product theory itself is bosonic, the condensation of $f$ generates a fermionic theory where $f$ is identified with the microscopic fermion in the quantum Hall system at $\nu = 1/2$.

To build a Lagrangian for the Moore-Read state, we begin with the product theory $U(2)_{2,-4} \times U(1)_8$ written in terms of a $U(2)$ gauge field $a$ and a $U(1)$ gauge field $b$:

$$L_{U(2)_{2,-4} \times U(1)_8} = -\frac{2}{4\pi} \text{Tr}\left(ada + \frac{2}{3}a^3\right) + \frac{2}{4\pi} \text{Tr}\, a\, d\, \text{Tr}\, a - \frac{8}{4\pi} b d b. \tag{A.9}$$

The one-form symmetry generated by $(j = 0, n = 2, m = 4)$ acts on the gauge fields as $a \to a + \lambda/2I, b \to b + \lambda/2$ where $\lambda$ is a properly normalized flat $U(1)$ gauge field. To implement the quotient, we therefore need to rewrite the Lagrangian in terms of new fields $c = a - bI$ and $\tilde{b} = 2b$ which are invariant under the action of the one-form symmetry. The resulting Lagrangian takes the form

$$L_{\text{MR}}[c, \tilde{b}] = -\frac{2}{4\pi} \text{Tr}\left(cdc + \frac{2}{3}c^3\right) + \frac{2}{4\pi} \text{Tr}\, c\, d\, \text{Tr}\, c + \frac{1}{2\pi} \tilde{b}\, d\, \text{Tr}\, c - \frac{1}{4\pi} \tilde{b} d \tilde{b}. \tag{A.10}$$

In order for this topological quantum field theory to describe the Moore-Read state, we need to appropriately couple it to the external electromagnetic $U(1)$ gauge field $A$ (technically a spin$_{\mathbb{C}}$ connection) through a mutual Chern-Simons term $\frac{1}{2\pi} A d \tilde{b}$. This coupling is fixed by the choice of the vison and ensures that the Moore-Read state has Hall conductance $\sigma_H = 1/2$. After including this coupling and integrating out the level-1 Chern-Simons term for $\tilde{b}$, we arrive at a simpler Lagrangian for the Moore-Read state:

$$L_{\text{MR}}[c] = -\frac{2}{4\pi} \text{Tr}\left(cdc + \frac{2}{3}c^3\right) + \frac{3}{4\pi} \text{Tr}\, c\, d\, \text{Tr}\, c + \frac{1}{2\pi} A d\, \text{Tr}\, c + \text{CS}[A, g], \tag{A.11}$$

where $\text{CS}[A, g] = \frac{1}{4\pi} A d A + 2\text{CS}_g$ is the usual Chern-Simons term for a spin$_{\mathbb{C}}$ connection. This identifies the TQFT of the Moore-Read state as $U(2)_{2,-8} \times U(1)_1$. It is easy to check that this Lagrangian reproduces the correct anyon data and Hall conductance of the Moore-Read state.

The above construction easily generalizes to other families of non-Abelian quantum Hall states. For example, if we replace the bosonic theory $U(2)_{2,-4}$ with $U(2)_{k,-2k}$, we produce the $k$-cluster Read-Rezayi states realized by fermions at filling $\nu = k/(k+2)$. The product theory $U(2)_{k,-2k} \times U(1)_{k(k+2)}$ contains a $\mathbb{Z}_k$ one-form symmetry generated by $(j = 0, n = 2, m = k+2)$. The resulting Read-Rezayi Lagrangian will map to $U(2)_{k,-2(k+2)} \times U(1)_1$. On the other hand, if we replace $U(1)_{k(k+2)}$ by $U(1)_{2k}$, the anyon $(j = 0, n = 2, m = k+2)$ has bosonic statistics and the quotient theory is a bosonic theory that maps to the bosonic Read-Rezayi states realized by bosons at filling $\nu = k/2$.

Finally, it is straightforward to introduce dispersive anyons into pure Chern-Simons theories by coupling in matter fields in various irreducible representations of $U(2)$. Given any parent $U(2)_{k_1, k_2}$ topological order, lattice translation acts projectively on the $(j, n)$ anyon as

$$T_x T_y T_x^{-1} T_y^{-1}[(j, n)] = e^{i\theta_{v,(j,n)}}[(j, n)], \tag{A.12}$$

where $\theta_{v,(j,n)}$ is the braiding phase between $(j, n)$ and the Abelian vison $v$ in the parent topological order. Define coprime integers $p_{(j,n)}, q_{(j,n)}$ such that $\theta_{v,(j,n)} = p_{(j,n)}/q_{(j,n)}$. The anyon

$(j, n)$ then moves in a $q_{(j,n)}$-fold reduced Brillouin zone. In a low-energy continuum description, the dispersion of $(j, n)$ can be captured by $q_{(j,n)}$ degenerate species of matter fields that transform projectively under the action of lattice translation. The complete Chern-Simons matter Lagrangian therefore takes the form

$$L = -\frac{k_1}{4\pi} \operatorname{Tr}\left( cdc + \frac{2}{3}c^3 \right) + \frac{2k_1 - k_2}{4} \frac{1}{4\pi} \operatorname{Tr} c \, d \operatorname{Tr} c + \sum_{\alpha=1}^{q_{(j,n)}} L_{(j,n)}[\Phi_\alpha, c], \quad (A.13)$$

where $L_{(j,n)}[\Phi_\alpha, c]$ describes a single matter field $\Phi_\alpha$ coupling to $c$ in the $(j, n)$ representation. For the Read-Rezayi states $\mathrm{RR}_k$ with arbitrary $k$, the appropriate Chern-Simons theory is $U(2)_{k,-2(k+2)}$ and the basic non-Abelian anyon is $(1/2, 1)$. The braiding phase between $(1/2, 1)$ and the vison is $2\pi/(k+2)$ and the electric charge of $(1/2, 1)$ is $e/(k+2)$. Therefore, the general CSGL Lagrangian for the $\mathrm{RR}_k$ state with $U(1)$ symmetry enrichment is

$$L = -\frac{k}{4\pi} \operatorname{Tr}\left( cdc + \frac{2}{3}c^3 \right) + \frac{k+1}{4\pi} \operatorname{Tr} c \, d \operatorname{Tr} c - \frac{1}{2\pi} \operatorname{Tr} c \, dA + \sum_{\alpha=1}^{k+2} L_{(1/2,1)}[\Phi_\alpha, c] + \mathrm{CS}[A, g]. \quad (A.14)$$

We use this Lagrangian as the starting point for analyzing the doped lattice Read-Rezayi states in Section 3 and Section 4.

### A.3 Composite fermion derivation of the $U(2)$ gauge theory for the Moore-Read state

Our description of the Moore-Read state as $U(2)_{2,-8} \times U(1)_1$ is rather abstract. Physically, it is often useful to think about the Moore-Read state more concretely as a $p + ip$ superconductor of composite fermions [26, 27]. The $h/(2e)$ vortex of the superconducting order parameter maps to the basic non-Abelian anyon $\sigma a$ and the neutral Bogoliubov quasiparticle maps to the Abelian anyon $\psi$. In this section, we follow this line of thinking and provide an explicit derivation of the $U(2)_{2,-8} \times U(1)_1$ Chern-Simons theory from the perspective of composite fermions.

In the standard composite fermion construction, we represent the microscopic electron $c$ as a composite fermion $f$ bound to a $4\pi$ flux of some fluctuating $U(1)$ gauge field $b$ [70, 71]. Instead of directly attaching flux, we write the microscopic electron operator as $c = f\phi$ where $\phi$ is a boson and $f$ is a composite fermion. This parton decomposition introduces a $U(1)$ gauge redundancy which can be removed by the introduction of a dynamical $U(1)$ gauge field $b$. The total Lagrangian can therefore be written as

$$L = L[f, A - b] + L[\phi, b]. \quad (A.15)$$

Using a mean-field ansatz in which $\nabla \times \boldsymbol{b} = \nabla \times \boldsymbol{A}$, the boson $\phi$ sees the microscopic magnetic flux while the composite fermion $f$ sees zero flux. At $\nu = 1/2$, the boson $\phi$ is at Landau level filling $\nu_\phi = 1/2$ and it is natural to put it into the $U(1)_2$ bosonic Laughlin state. The resulting Lagrangian is

$$L = L[f, A - b] - \frac{2}{4\pi} \beta d\beta + \frac{1}{2\pi} \beta db. \quad (A.16)$$

In this Lagrangian, all the Chern-Simons terms are properly quantized to be integers. Naively integrating out $\beta$ recovers the Lagrangian in (A.15) but breaks gauge-invariance. Therefore, we will keep the $\beta$ gauge field intact and proceed.

Following the constructions in Refs. [26, 27], we anticipate that the Moore-Read state can be obtained by putting $f$ into a $p + ip$ topological superconductor state. Such a topological superconductor is defined by the following universal features: (1) it spontaneously breaks the

$U(1)$ symmetry down to $\mathbb{Z}_2$; (2) it has a chiral central charge $c_- = 1/2$; (3) it is an invertible phase with no intrinsic topological order [34]. We claim that such a phase can be described by the following $U(2)_{2,0} \times U(1)_{-1}$ Chern-Simons theory:

$$L_{p+ip}[f,a] = -\frac{2}{4\pi} \text{Tr}\left( cdc + \frac{2}{3}c^3 \right) + \frac{1}{4\pi} \text{Tr}\, c \, d \, \text{Tr}\, c + \frac{1}{2\pi} \text{Tr}\, c \, da - \text{CS}[a,g], \qquad \text{(A.17)}$$

where $c$ is a $U(2)$ gauge field and $\text{CS}[a,g]$ is the usual Chern-Simons term for the $\text{spin}_{\mathbb{C}}$ connection $a$ which describes the response theory for $U(1)_{-1}$. The mutual Chern-Simons term between $\text{Tr}\, c$ and $a$ Higgses $U(1)$ down to a $\mathbb{Z}_2$ subgroup, implying the formation of a charge-2 superconductor. The chiral central charge of the $U(2)_{2,0}$ theory is $3/2$, while the chiral central charge of the decoupled $U(1)_{-1}$ sector is $-1$. Therefore, the total chiral central charge is $c_- = 1/2$, again in agreement with the $p+ip$ superconductor. Finally, using the quotient representation $U(2)_{2,0} = SU(2)_2 \times U(1)_0/\mathbb{Z}_2$, we know that $U(2)_{2,0}$ has total squared quantum dimension $4/2 = 2$. Since $U(2)_{2,0}$ is a fermionic theory, this squared quantum dimension is saturated by $\{1, c\}$ where $c$ is the microscopic electron. Hence, $U(2)_{2,0}$ contains no intrinsic topological order. This completes the matching between $U(2)_{2,0} \times U(1)_{-1}$ and the $p+ip$ topological superconductor.

We can now combine the $U(2)_{2,0} \times U(1)_{-1}$ description of the $p+ip$ superconductor formed by $f$ with the $U(1)_2$ state formed by $\phi$. The total effective Lagrangian takes the form

$$
\begin{aligned}
L_{\text{eff}} = & -\frac{2}{4\pi} \text{Tr}\left( cdc + \frac{2}{3}c^3 \right) + \frac{1}{4\pi} \text{Tr}\, c \, d \, \text{Tr}\, c + \frac{1}{2\pi} \text{Tr}\, c \, d(A-b) \\
& -\frac{1}{4\pi}(A-b)d(A-b) - 2\text{CS}_g - \frac{2}{4\pi}\beta d\beta + \frac{1}{2\pi}\beta db.
\end{aligned}
\qquad \text{(A.18)}
$$

The self Chern-Simons level for $b$ is $1$. Therefore, $b$ can be integrated out and the effective Lagrangian reduces to

$$
\begin{aligned}
L_{\text{eff}} = & -\frac{2}{4\pi} \text{Tr}\left( cdc + \frac{2}{3}c^3 \right) + \frac{1}{4\pi} \text{Tr}\, c \, d \, \text{Tr}\, c + \frac{1}{2\pi} \text{Tr}\, c \, dA - \frac{2}{4\pi}\beta d\beta \\
& -\frac{1}{4\pi}AdA - 2\text{CS}_g + \frac{1}{4\pi}(A - \text{Tr}\, c + \beta)d(A - \text{Tr}\, c + \beta) + 2\text{CS}_g \\
= & -\frac{2}{4\pi} \text{Tr}\left( cdc + \frac{2}{3}c^3 \right) + \frac{2}{4\pi} \text{Tr}\, c \, d \, \text{Tr}\, c - \frac{1}{4\pi}\beta d\beta + \frac{1}{2\pi}\beta d(A - \text{Tr}\, c).
\end{aligned}
\qquad \text{(A.19)}
$$

Now the total Chern-Simons level for $\beta$ is also $1$ and $\beta$ can be integrated out as well. The final theory written in terms of the $U(2)$ gauge field $c$ takes the form

$$L_{\text{eff}} = -\frac{2}{4\pi} \text{Tr}\left( cdc + \frac{2}{3}c^3 \right) + \frac{3}{4\pi} \text{Tr}\, c \, d \, \text{Tr}\, c - \frac{1}{2\pi} \text{Tr}\, c \, dA + \frac{1}{4\pi}AdA + 2\text{CS}_g. \qquad \text{(A.20)}$$

This is precisely the canonical Lagrangian for the $U(2)_{2,-8} \times U(1)_1$ non-Abelian Chern-Simons theory which describes the Moore-Read state.

# B  Dynamics of non-Abelian Maxwell Chern-Simons theories

In this Appendix, we study dynamical properties of Maxwell Chern-Simons (MCS) theories coupled to massive scalar fields. The fundamental excitations of this theory are anyons sourced by the scalar fields. The overarching goal is to understand the effective interactions between these anyons mediated by the non-Abelian gauge fields. These interactions vanish in pure Chern-Simons theories where the gauge fields have infinite mass. Therefore, it is necessary to

introduce a Maxwell term that renders the photon mass finite. We can regard this Maxwell term as the leading correction to the pure Chern-Simons theory generated by integrating out high-energy degrees of freedom. In what follows, we will first set up our conventions for a general $U(2)$ MCS theory and then extract the effective interactions.

We begin by defining the Lagrangian for the $U(2)_{k_1,k_2}$ MCS theory:

$$L = -\frac{k_1}{4\pi}\mathrm{Tr}\left(cdc + \frac{2}{3}c^3\right) + \frac{k_2 + 2k_1}{4}\frac{1}{4\pi}\mathrm{Tr}\,cd\,\mathrm{Tr}\,c + \frac{1}{2g^2}\mathrm{Tr}\,f \wedge \star f\,, \tag{B.1}$$

where the non-Abelian field strength is related to the gauge field $c$ through the standard relation $f = dc + c^2$. For the purpose of extracting effective interactions, the topology of the gauge fields does not matter and we can decouple the Euclidean Lagrangian into a $U(1)$ part and an $SU(2)$ part:

$$\begin{aligned}
L_{U(1)} &= -\frac{ik_2}{4\cdot 4\pi}c_0 dc_0 + \frac{1}{4g^2}dc_0 \wedge \star dc_0\,,\\
L_{SU(2)} &= -\frac{ik_1}{4\pi}\mathrm{Tr}\left(cdc + \frac{2}{3}c^3\right) + \frac{1}{2g^2}\mathrm{Tr}\left(dc + c^2\right) \wedge \star(dc + c^2)\,,
\end{aligned} \tag{B.2}$$

where we used the Lie algebra decomposition $c = c_0 I/2 + c_a \sigma^a/2$ with $\sigma^a$ the Pauli matrices. The Abelian sector is a Gaussian theory that can be exactly solved. The non-Abelian sector is interacting and does not admit an exact solution. However, in the weak-coupling regime where $g^2$ is much smaller than the mass in the matter sector, perturbation theory is well-behaved and self-interactions between the gauge fields are suppressed at low energies. Therefore, it is reasonable to estimate the gauge field propagator using the Gaussian approximation. Below, we will study the $U(1)$ and $SU(2)$ sectors in order.

## B.1 Abelian sector: $U(1)_k$

We first consider the Abelian sector described by a Euclidean action

$$S = \int d^3x\,\frac{1}{4g^2}f_{\mu\nu}^2 + \frac{ik}{4\pi}\epsilon^{\mu\lambda\nu}a_\mu\partial_\lambda a_\nu = \frac{1}{2}\int \frac{d^3q}{(2\pi)^3}a^\mu(-q)D_{\mu\nu}^{-1}(q)a^\nu(q)\,, \tag{B.3}$$

where the inverse propagator $D_{\mu\nu}^{-1}$ can be written in the $R_\xi$ gauge as

$$D_{\mu\nu}^{-1}(q) = \frac{1}{g^2}\left[q^2 g_{\mu\nu} - (1 - \xi^{-1})q_\mu q_\nu\right] + i\frac{k}{2\pi}\epsilon_{\mu\lambda\nu}(iq^\lambda)\,. \tag{B.4}$$

We define the kernel $L_{\mu\nu} = \epsilon_{\mu\lambda\nu}(iq^\lambda)$, which satisfies the following identities:

$$L_{\mu\nu}^2 = \epsilon_{\mu\lambda\alpha}(iq^\lambda)\epsilon_{\alpha\gamma\nu}(iq^\gamma) = g_{\mu\nu}q^2 - q_\mu q_\nu\,,\quad L_{\mu\nu}^3 = q^2 L_{\mu\nu}\,. \tag{B.5}$$

A general two-index tensor of the form $AL_{\mu\nu} + B\left[L_{\mu\nu}^2 + \xi^{-1}q_\mu q_\nu\right]$ can be inverted as

$$D_{\mu\nu} = CL_{\mu\alpha} + D\left[L_{\mu\alpha}^2 + f(\xi)q_\mu q_\alpha\right]\,, \tag{B.6}$$

where the constants $C, D, f(\xi)$ satisfy

$$C = \frac{Aq^{-2}}{A^2 - B^2 q^2}\,,\quad D = \frac{-Bq^{-2}}{A^2 - B^2 q^2}\,,\quad f(\xi) = \frac{B^2 q^2 - A^2}{B^2 q^2}\xi\,. \tag{B.7}$$

Focusing on the case of interest to us $A = \frac{ik}{2\pi}$ and $B = \frac{1}{g^2}$, we find

$$\langle a_\mu(q) a_\nu(q) \rangle = \frac{-ik/(2\pi)}{q^2(\frac{k^2}{4\pi^2} + \frac{q^2}{g^4})} \epsilon_{\mu\lambda\nu}(iq_\lambda) + \frac{1/g^2}{q^2(\frac{k^2}{4\pi^2} + \frac{q^2}{g^4})} \left(g_{\mu\nu}q^2 - q_\mu q_\nu\right) + \frac{g^2\xi}{q^4} q_\mu q_\nu. \quad \text{(B.8)}$$

Defining the photon mass $M = \frac{g^2 k}{2\pi}$, we can write this in a more compact form

$$\begin{aligned}
D_{\mu\nu}(q) \equiv \langle a_\mu(q) a_\nu(q) \rangle &= \frac{-ig^2 M}{q^2(M^2 + q^2)} \epsilon_{\mu\lambda\nu}(iq_\lambda) + \frac{g^2}{q^2(M^2 + q^2)} \left(g_{\mu\nu}q^2 - q_\mu q_\nu\right) \\
&\quad + \frac{g^2\xi}{q^4} q_\mu q_\nu.
\end{aligned} \quad \text{(B.9)}$$

This propagator is analytic in the complex $p$ plane except for poles at $p = \pm iM$ and $p = 0$. The pole at $p = 0$ is gauge-dependent and not physical. The physical pole at $p = \pm iM$ indicates the existence of a massive mode with mass gap $M$. This is the familiar phenomenon of topological mass generation in MCS theories.

Given the gauge propagator $D_{\mu\nu}$, we can integrate out the gauge fields and generate an effective interaction between the matter field currents $J^\mu$:

$$\begin{aligned}
Z &= \int Da \exp\left\{ -\frac{1}{2} \int d^3x \left[ a^T D^{-1} a - a^T J - J^T a \right] \right\} \\
&= \int Da \exp\left\{ -\frac{1}{2} \int d^3x \left[ (a^T - J^T D) D^{-1} (a - DJ) \right] + \frac{1}{2} \int d^3x J^T D J \right\} \\
&= \int D\tilde{a} \exp\left\{ -\frac{1}{2} \int d^3x \left[ \tilde{a}^T D^{-1} \tilde{a} \right] + \frac{1}{2} \int d^3x J^T D J \right\}.
\end{aligned} \quad \text{(B.10)}$$

For massive scalar fields, we can restrict to static density-density interactions that correspond to $J^0(\omega = 0, q) D_{00}(\omega = 0, \boldsymbol{q}) J^0(\omega = 0, -q)$. Fourier transforming $D_{00}(\omega = 0, \boldsymbol{q})$ to position space, we obtain

$$D_{00}(\omega = 0, \boldsymbol{x}) = \int \frac{d^2\boldsymbol{q}}{(2\pi)^2} \frac{g^2 e^{iq\cdot x}}{M^2 + |\boldsymbol{q}|^2} = \frac{g^2}{2\pi} K_0(M|\boldsymbol{x}|) \xrightarrow{|\boldsymbol{x}|\to\infty} \frac{g^2}{\sqrt{8\pi M|\boldsymbol{x}|}} e^{-M|\boldsymbol{x}|}. \quad \text{(B.11)}$$

As expected, the topological mass $M$ generated by the Chern-Simons term screens the long-range Coulomb interactions of pure Maxwell theory to a short-range interaction with decay length set by $1/M$. The sign structure is such that the interaction between anyons with gauge charges of the same (opposite) sign is repulsive (attractive).

## B.2 Non-Abelian sector: $SU(2)_k$

The above considerations generalize easily to non-Abelian MCS theory. Let us consider the basic example $SU(2)_k$. At tree-level, we can neglect the non-linear gauge field self-interactions and focus on the Gaussian part of the gauge field Lagrangian. Choosing the convention $T^a = \sigma^a/2$ and working in the $R_\xi$ gauge (at tree level, we do not need to worry about the interactions between ghosts and the gauge field), this Lagrangian can be simplified to

$$\begin{aligned}
L_{\text{Gaussian}} &= \frac{ik}{4\pi} \text{Tr}\, cdc + \frac{1}{2g^2} \text{Tr}\, dc \wedge \star dc = \frac{ik}{4\pi} \epsilon^{\mu\lambda\nu} c_\mu^a \partial_\lambda c_\nu^b \, \text{Tr}\, T_a T_b \\
&\quad + \frac{1}{4g^2} (\partial_\mu c_\nu^a - \partial_\nu c_\mu^a)(\partial_\mu c_\nu^b - \partial_\nu c_\mu^b) \, \text{Tr}\, T_a T_b \\
&= \frac{ik}{8\pi} c_\mu^a \epsilon^{\mu\lambda\nu}(iq_\lambda) c_\nu^b \delta_{ab} + \frac{1}{4g^2} c_\mu^a \delta_{ab} \left[ q^2 g^{\mu\nu} - (1 - \xi^{-1}) q^\mu q^\nu \right] c_\nu^b.
\end{aligned} \quad \text{(B.12)}$$

From this Lagrangian, we can read off the inverse gauge propagator

$$\left(D^{-1}\right)^{\mu\nu}_{ab} = \delta_{ab}\left(\frac{ik}{4\pi}\epsilon^{\mu\lambda\nu}iq_\lambda + \frac{1}{2g^2}\left[q^2 g^{\mu\nu} - (1-\xi^{-1})q^\mu q^\nu\right]\right) = \frac{1}{2}\delta_{ab}\bar{D}^{-1}_{\mu\nu}, \tag{B.13}$$

where $\bar{D}_{\mu\nu}$ is the Abelian propagator. Using our results from the Abelian case, we can immediately write down the non-Abelian gauge propagator

$$\begin{aligned}
D^{ab}_{\mu\nu} &= 2\delta^{ab}\bar{D}_{\mu\nu}\\
&= 2\delta^{ab}\left[\frac{-ig^2 M}{q^2(M^2+q^2)}\epsilon_{\mu\nu\lambda}(iq_\lambda) + \frac{g^2}{q^2(M^2+q^2)}(g_{\mu\nu}q^2 - q_\mu q_\nu) + \frac{g^2\xi}{q^4}q_\mu q_\nu\right].
\end{aligned} \tag{B.14}$$

The leading-order non-Abelian current-current interaction therefore takes the form

$$L_{\text{int}} = \frac{1}{2}J^\mu_a D^{ab}_{\mu\nu}J^\nu_b = \sum_a J^\mu_a D_{\mu\nu}J^\nu_a, \quad J^\mu_a = \left[\Phi^*_i(i\partial^\mu\Phi_j) - (i\partial^\mu\Phi^*_i)\Phi_j\right]T^{ij}_a. \tag{B.15}$$

Expanding the current-current interactions in terms of fundamental scalar fields, we find that

$$L_{\text{int}} = \sum_a \left[\Phi^*_i(i\partial^\mu\Phi_j) - (i\partial^\mu\Phi^*_i)\Phi_j\right](T^a_{ij}D_{\mu\nu}T^a_{kl})\left[\Phi^*_k(i\partial^\nu\Phi_l) - (i\partial^\nu\Phi^*_k)\Phi_l\right]. \tag{B.16}$$

Since the physical poles of $D_{\mu\nu}$ are located at $p = \pm iM$, the non-Abelian gauge fields mediate short-range interactions between any pair of scalar fields transforming in the spin $j_1, j_2$ representations of $SU(2)$. However, unlike in the Abelian case, $j_1$ and $j_2$ can fuse into multiple different channels and the sign structure of the interactions can be different for distinct fusion channels. The simplest non-trivial example relevant for the non-Abelian quantum Hall states is where $j_1 = j_2 = 1/2$. The fusion rule is $1/2 \otimes 1/2 = 0 \oplus 1$. Therefore, we need to study the projected interactions in the singlet and triplet channels. To that end, we define the index-resolved interaction form factor

$$V_{ik,jl} = \sum_a T^a_{ij}T^a_{kl}, \tag{B.17}$$

where $i, k$ are associated with scalar creation operators while $j, l$ are associated with scalar annihilation operators. The projection operators onto the singlet and triplet sectors are

$$P^{(0)}_{ik,jl} = \frac{1}{2}\epsilon_{ik}\epsilon_{jl}, \quad P^{(1)}_{ik,jl} = \delta_{ij}\delta_{kl} - P^{(0)}_{ik,jl}. \tag{B.18}$$

We can check that $P^{(0)}$ satisfies the definition of a projector

$$\sum_{kl} P^{(0)}_{ij,kl}P^{(0)}_{kl,mn} = \frac{1}{4}\sum_{kl}\epsilon_{ij}\epsilon_{kl}\epsilon_{kl}\epsilon_{mn} = \frac{1}{2}\epsilon_{ij}\epsilon_{mn} = P^{(0)}_{ij,mn}. \tag{B.19}$$

Now we can project the interaction form factor $V_{ij,kl}$ onto the singlet and triplet channels

$$\begin{aligned}
V^{(0)} &= \text{Tr}\,VP^{(0)} = \frac{1}{2}\sum_{ijkl}\epsilon_{ij}\epsilon_{kl}\sum_a T^a_{ki}T^a_{lj} = \frac{1}{8}\sum_{ik}(2\epsilon_{ik}\epsilon_{ki} - \epsilon^2_{ik}) = -\frac{3}{4},\\
V^{(1)} &= \text{Tr}\,VP^{(1)} = \frac{3}{4} + \sum_{ijkl}\sum_a T^a_{ik}T^a_{jl}\delta_{ki}\delta_{lj} = \frac{3}{4} + \sum_a(\text{Tr}\,T^a)^2 = \frac{3}{4}.
\end{aligned} \tag{B.20}$$

The important takeaway is that $SU(2)$ gauge fields in the $SU(2)_k$ MCS theory mediate attraction in the singlet channel but repulsion in the triplet channel. This means that the $j = 0$ singlet is energetically favored over the $j = 1$ triplet when a pair of $j = 1/2$ quasiparticles fuse. This simple result is invoked in the physical picture of Section 3.

# C  Monopole operators of the $U(N)_{k_1,k_2}$ non-Abelian Chern-Simons theory

In this Appendix, we review the construction of gauge-invariant monopole operators of a general $U(N)_{k_1,k_2}$ non-Abelian Chern-Simons theory coupled to $m$ species of matter fields each in the fundamental representation of $U(N)$. The Lagrangian for this theory takes the form

$$L = -\frac{k_1}{4\pi}\mathrm{Tr}\left(cdc + \frac{2}{3}c^3\right) + \frac{2k_1 - k_2}{4}\frac{1}{4\pi}\mathrm{Tr}\,cd\,\mathrm{Tr}\,c + \frac{1}{2\pi}Ad\,\mathrm{Tr}\,c + \sum_{\alpha=1}^{n} L[\Phi_\alpha, c]. \tag{C.1}$$

This non-Abelian gauge theory has a single $U(1)$ topological current which couples to the external electromagnetic gauge field $A$:

$$J^{\mathrm{top}} = \frac{1}{2\pi}\mathrm{Tr}\,F = \frac{1}{2\pi}\mathrm{Tr}(dc + c \wedge c) = \frac{1}{2\pi}d\,\mathrm{Tr}\,c. \tag{C.2}$$

For all $N$, this topological current is quantized to be an integer. This is sometimes referred to as the topological charge. To define a classical monopole at the origin, we cover the sphere surrounding the origin with two overlapping coordinate charts and define a gauge field configuration

$$c = \frac{1}{2}H\begin{cases}(1 - \cos\theta)d\phi, & \theta \neq \pi\,\mathrm{chart}, \\ -1 - (\cos\theta)d\phi, & \theta \neq 0\,\mathrm{chart},\end{cases} \tag{C.3}$$

where $H$ is a constant Hermitian $N \times N$ matrix with eigenvalues $\{q_1, \ldots, q_N\}$. These eigenvalues are the GNO charges [72] associated with the Cartan subalgebra $U(1)^N \subset U(N)$.

A quantization condition on the eigenvalues can be derived from gauge invariance. Recall that for any closed two-cycle $S$, the field strength profile $F = \frac{q}{2}\sin\theta d\theta \wedge d\phi$ integrates to $\int_S F = 2\pi q$. On the overlap between the two coordinate charts, the two definitions of $c$ have to agree up to a $U(N)$ gauge transformation. Using the field strength computed from $c$, we see that this is the case if and only if $e^{2\pi iH} = 1$, which gives the quantization condition $q_i \in \mathbb{Z}$ for all $i$. Note that the individual GNO charges $q_i$ are not conserved by the dynamics of the theory. Instead, the only conserved charge is the topological charge $Q^{\mathrm{top}} = \sum_{i=1}^{N} q_i$.

In $U(N)$ Yang-Mills theories, the bare monopole operators $\mathcal{M}_{\{q_i\}}$ are gauge-invariant. However, in $U(N)$ Chern-Simons matter theories, the same monopoles are charged under the $U(2)$ gauge field and do not represent gauge-invariant operators. To obtain gauge-invariant monopoles, we need to dress the bare monopole by matter fields with the opposite gauge charge. The goal of our analysis is to identify the spectrum of gauge-invariant monopoles when the Lagrangian contains a particular matter field. In principle, this analysis can be done for any matter field that transforms in some representation of $U(N)$. However, we will focus on the fundamental representation of $U(2)$, because it illustrates the essential principles and is most relevant for the CSGL theories considered in the main text.

With the preceding general discussion in mind, let us analyze the $U(2)_{k_1,k_2}$ Chern-Simons theory with $m$ matter fields transforming in the fundamental representation of $U(2)$. We begin by considering the simplest bare monopole operator with $(q_1 = 1, q_2 = 0)$. Decomposing $U(2)$ as $SU(2) \times U(1)/\mathbb{Z}_2$, we see that this monopole has half flux in the $U(1)$ direction and half flux in the $SU(2)$ direction. Since the $SU(2)$ CS level is $k_1$ and the $U(1)$ CS level is $k_2$, the monopole annihilation operator carries charge $-k_2/2$ under $U(1)$ and spin $-k_1/2$ under $SU(2)$. We remind the reader that $k_2$ is necessarily even due to the condition $2k_1 + k_2 \in 4\mathbb{Z}$. In other words, the monopole lives in the $(j = -k_1/2, n = -k_2/2)$ representation of $U(2)$. Since this representation is $k_1 + 1$ dimensional, we conclude that the bare monopole in fact carries an internal index $a = 1, \ldots, k_1 + 1$ and can be labeled as $\mathcal{M}^a_{(1,0)}$.

To construct the gauge-invariant monopole, we need to insert composite operators built out of the matter fields $\Phi_\alpha$, each transforming in the $(j = k_1/2, n = k_2/2)$ representation of $U(2)$, thereby canceling the gauge charge of $\mathcal{M}^a_{(1,0)}$. A matter field of the form $\left(\text{Sym}^{k_1}\Phi^*\right)_a$ carries the representation $(j = k_1/2, n = k_1)$, which has the correct value of $j$ but the incorrect value of $n$. To get the correct value of $n$, we need to multiply $\left(\text{Sym}^{k_1}\Phi^*\right)_a$ by another operator with $(j = 0, n = -k_1 + k_2/2)$. Such $SU(2)$-invariant operators can be built from elementary baryon operators of the form:

$$B_{\alpha\beta} = \epsilon_{ij}\Phi^i_\alpha \Phi^j_\beta \,, \tag{C.4}$$

where $\alpha, \beta \in \{1, 2, \ldots, m\}$. Since the baryon is quadratic in $\Phi_\alpha$, it carries $U(1)$ charge $-2$. Therefore, $B^{k_1/2-k_2/4}_{\alpha\beta}$ carries the correct representation $(j = 0, n = -k_1 + k_2/2)$. Note that this expression makes sense only when $k_1/2 - k_2/4$ is an integer. This is always true as can be inferred from the two consistency conditions for $U(2)_{k_1, k_2}$ that we derived in Appendix A:

$$2k_1 + k_2 = 0 \mod 4\,, \quad k_2 = 0 \mod 2 \quad \rightarrow \quad 2k_1 - k_2 = 2k_2 \mod 4 = 0\,. \tag{C.5}$$

Putting everything together, we conclude that the elementary gauge-invariant monopole with $q_1 = 1, q_2 = 0$ can be written schematically as

$$\tilde{\mathcal{M}}_{(1,0)} \sim \sum_{a=1}^{k_1+1} \mathcal{M}^a_{(1,0)} \cdot \left(\text{Sym}^{k_1}\Phi^*\right)_a \cdot B^{(2k_1-k_2)/4}\,. \tag{C.6}$$

Let us pause to make two remarks about this expression. The first remark is that $\tilde{\mathcal{M}}_{(1,0)}$ is not a single operator but rather a family of operators, as each power of $B_{\alpha\beta}$ in $B^{(2k_1-k_2)/4}$ can carry arbitrary $\alpha, \beta$ indices. The schematic form in (C.6) keeps track of the $U(2)$ color indices but not the matter field species index. The second remark is that the family of monopoles in (C.6) does not exhaust all monopoles in the $(q_1 = 1, q_2 = 0)$ flux sector. One can always generate monopoles in the same flux sector by attaching additional neutral operators such as $|\Phi_\alpha|^2$ or adding derivatives. This family is nevertheless special as it contains monopoles with the minimum number of matter fields and spacetime derivatives.

A similar construction applies to the flux sector $q_1 = 0, q_2 = 1$. The monopole annihilation operator now carries charge $-k_2/2$ under $U(1)$ and $j = k_1/2$ under $SU(2)$. To construct the gauge-invariant monopole, we need to insert composite operators built out of the matter fields $\Phi_\alpha$, each transforming in the $(j = -k_1/2, n = k_2/2)$ representation of $U(2)$. A matter field of the form $\left(\text{Sym}^{k_1}\Phi\right)_a$ carries the representation $(j = -k_1/2, n = -k_1)$, which has the correct value of $j$ but the incorrect value of $n$. To get the correct value of $n$, we need to multiply by another field in the representation $(j = 0, n = k_2/2 + k_1)$, which maps to a baryon operator $B^{-k_1/2-k_2/4}_{\alpha\beta}$. Therefore, the elementary gauge-invariant monopole with $q_1 = 0, q_2 = 1$ can be written as

$$\tilde{\mathcal{M}}_{(0,1)} \sim \sum_{a=1}^{k_1+1} \mathcal{M}^a_{(0,1)} \cdot \left(\text{Sym}^{k_1}\Phi\right)_a \cdot B^{-(2k_1+k_2)/4}\,. \tag{C.7}$$

We now move on to the flux sector $q_1 = 1, q_2 = 1$. Now, the monopole carries a unit flux in the $U(1)$ direction, but zero flux in the $SU(2)$ direction. The Chern-Simons terms then imply that the monopole lives in the $(j = 0, n = -k_2)$ representation of $U(2)$. Unlike the $(q_1 = 1, q_2 = 0)$ monopole, the absence of an $SU(2)$ charge implies that this monopole has no internal index and can be labeled simply as $\mathcal{M}_{(1,1)}$. To construct the gauge-invariant operator, we need to attach a composite operator built from the matter fields that transforms in the $(j = 0, n = k_2)$ representation. Using the definition of the baryon operators and the fact that $k_2 = 0 \mod 2$, we see that the simplest composite operator with $(j = 0, n = k_2)$ is $B^{-k_2/2}$. The minimal gauge-invariant monopole is therefore

$$\tilde{\mathcal{M}}_{(1,1)} \sim \mathcal{M}_{(1,1)}B^{-k_2/2}\,. \tag{C.8}$$

This analysis generalizes to monopoles with arbitrary flux assignment $(q_1, \ldots, q_N)$ in $U(N)_{k_1,k_2}$. Since the physical phenomena of interest to us only involve monopoles with small total flux $q_1 + q_2$ in the $N = 2$ theory, we will not pursue a fully general discussion here.

## D  Pairing of bosonic CFLs in the $\nu_{\text{tot}} = 1 + 1$ bosonic quantum Hall bilayer

In this Appendix, we study the $\nu = 1+1$ bosonic quantum Hall bilayer with a global $U(1) \times U(1)$ symmetry corresponding to particle number conservation in each layer (this setup was previously considered in an unpublished work [73]). We begin in the decoupled limit, where each layer forms a bosonic composite Fermi liquid state. The corresponding Lagrangian takes the form

$$L_{12} = L_{\text{FL}}[f_1, a_1] + L_{\text{FL}}[f_2, a_2] + \frac{1}{4\pi}(a_1 - A_1)d(a_1 - A_1) + \frac{1}{4\pi}(a_2 - A_2)d(a_2 - A_2) + 4\text{CS}_g \,, \quad \text{(D.1)}$$

where $f_1, f_2$ are the composite fermions and $A_1, A_2$ are the external gauge fields corresponding to the $U(1) \times U(1)$ global symmetry.

When interlayer interactions are turned on, a renormalization group analysis shows that fluctuations of $a_1$ and $a_2$ favor BCS pairing between the two composite fermion species [74]. Following the intuition from fermionic CFL bilayers, we assume that the dominant pairing channel is $p + ip$ [75–79]. The paired state has $\nu_{\text{Kitaev}} = 2$ in the Kitaev 16-fold way and is described by the TQFT

$$L[f_1, a_1] + L[f_2, a_2] \to \frac{1}{4\pi}(\beta_1 - \beta_2)d(\beta_1 - \beta_2) + \frac{1}{2\pi}a_1 d\beta_1 + \frac{1}{2\pi}a_2 d\beta_2 \,. \quad \text{(D.2)}$$

Gluing this TQFT back to the remaining terms in $L_{12}$, we find

$$\begin{aligned}
L_{12} = {} & \frac{1}{4\pi}(a_1 - A_1)d(a_1 - A_1) + \frac{1}{4\pi}(a_2 - A_2)d(a_2 - A_2) + 4\text{CS}_g \\
& + \frac{1}{4\pi}(\beta_1 - \beta_2)d(\beta_1 - \beta_2) + \frac{1}{2\pi}a_1 d\beta_1 + \frac{1}{2\pi}a_2 d\beta_2 \,.
\end{aligned} \quad \text{(D.3)}$$

The equations of motion for $a_i$ set $a_i = A_i - \beta_i$. Integrating out $a_i$ therefore generates

$$\begin{aligned}
L_{12} = {} & \frac{1}{4\pi}\beta_1 d\beta_1 + \frac{1}{4\pi}\beta_2 d\beta_2 + \frac{1}{4\pi}(\beta_1 - \beta_2)d(\beta_1 - \beta_2) \\
& + \frac{1}{2\pi}(A_1 - \beta_1)d\beta_1 + \frac{1}{2\pi}(A_2 - \beta_2)d\beta_2 \\
= {} & -\frac{1}{2\pi}\beta_1 d\beta_2 + \frac{1}{2\pi}A_1 d\beta_1 + \frac{1}{2\pi}A_2 d\beta_2 \,.
\end{aligned} \quad \text{(D.4)}$$

We recognize this final Lagrangian as the Lagrangian for a bosonic IQH state [46]. Therefore, $p + ip$ pairing between two bosonic CFLs produces an invertible gapped phase. When $A_1 = A_2 = A$, integrating out $\beta_1$ and $\beta_2$ generates a simple Chern-Simons response

$$L_{12}[A] = \frac{2}{4\pi}A dA \,. \quad \text{(D.5)}$$

This is the result invoked in Section 4.

# E  Minimal modular extensions of the Pfaffian state

In this Appendix, we describe in more detail the minimal modular extensions of the Moore-Read Pfaffian state. The minimal modular extensions of a fermionic theory are the theories obtained by gauging fermion parity (summing over spin structures), and are also known as bosonic shadows/parents in the literature. They are obtained from the original fermionic theory by adding in new quasiparticles (fermion parity fluxes) that braid with phase $-1$ with the fermion. These theories are used in Sec. 5.2 to interpret the field theory result in terms of charge/flux unbinding. The Pfaffian state is interesting from the perspective of fermionic topological orders because it does not factorize into a bosonic theory with a trivial fermionic theory, unlike all Abelian fermionic topological orders and many non-Abelian ones [48, 80]. The anti-Pfaffian and PH-Pfaffian states also do not factorize. Therefore, the minimal modular extensions cannot just be obtained from a bosonic theory stacked with the modular extensions of the trivial fermionic theory $\{1, f\}$ given in [34]. In general, finding the complete algebraic data determining a minimal modular extensions of a fermionic topological orders that do not factorize is a difficult problem without a general solution. In this case, it is possible to find these extensions because we can simply undo the $/\mathbb{Z}_2$ used to define the Pfaffian state. We will show here the anyon data of the minimal modular extensions of the Pfaffian state with $c_- = 3/2$ and $c_- = 2$; the others can be derived in a similar way.

The most obvious minimal modular extension of the Pfaffian state is simply Ising $\times U(1)_8$. The fermion is still identified with $\psi a^4$ and the fermion parity vortices include $\sigma$ and $a$; these anyons were projected out by the $/\mathbb{Z}_2$ to get the Pfaffian state. The chiral central charge is $c_- = 3/2$.

Other minimal modular extensions are obtained by stacking with copies of Ising and condensing a pair of fermions. We do not need to be concerned about enrichment with a $U(1)$ symmetry because the modular extension corresponds to the stacked neutral theory in Sec. 5.2. For example, the $c_- = 2$ minimal modular extension is obtained by stacking with Ising and condensing $\psi_1 \psi_2 a^4$ (note that we do not condense $\psi_1 \psi_2$ because the fermion of interest in the $c_- = 3/2$ modular extension is identified with $\psi_1 a^4$) where $\psi_1$ comes from the original Pfaffian state and $\psi_2$ comes from the stacked Ising theory. This removes anyons like $\sigma_1$ and $a$ because they braid nontrivially with $\psi_1 \psi_2 a^4$, and identifies $\psi_1 a^4 \sim \psi_2$. The anyon $\sigma_2 a$ remains deconfined and serves as a fermion parity flux. One might suspect that $\sigma_1 \sigma_2$ might also be a fermion parity flux. However,

$$\sigma_1 \sigma_2 \times \sigma_1 \sigma_2 = 1 + \psi_1 + \psi_2 + \psi_1 \psi_2 = 1 + \psi_2 a^4 + \psi_2 + a^4. \tag{E.1}$$

Fusing in $a^4$ gives $2 + 2\psi_2$, which includes two copies of the vacuum. This indicates an inconsistency. Since $\sigma_1 \sigma_2 a^2$ fuses with itself to give two copies of the vacuum, it actually splits into two Abelian anyons. Let us call these anyons $e$ and $m$ with $e = \psi_2 m$, i.e. $\sigma_1 \sigma_2 a^2 \sim e + m$ and $\sigma_1 \sigma_2 = (e + m)a^6$. $e$ and $m$ have topological spin $3/8$, braid with $\psi_1$ with a $-1$ phase, and satisfy $e^2 = m^2 = 1$. The 24 anyons types are given by

$$\begin{aligned} \{ &1, a^2, a^4, a^6, \sigma_1 a, \sigma_1 a^3, \sigma_1 a^5, \sigma_1 a^7, \psi_1, \psi_1 a^2, \psi_1 a^4, \psi_1 a^6, \\ &\sigma_2 a, \sigma_2 a^3, \sigma_2 a^5, \sigma_2 a^7, e, m, ea^2, ma^2, ea^4, ma^4, ea^6, ma^6 \}, \end{aligned} \tag{E.2}$$

with fusion rules that include

$$\sigma_1 a \times \sigma_2 a = e + m, \qquad e \times \sigma_2 a = \sigma_1 a^3, \qquad m \times \sigma_2 a = \psi_2 \sigma_1 a^3 = \sigma_1 a^7. \tag{E.3}$$

The other 14 minimal modular extensions are obtained by sequentially stacking with Ising and condensing fermion-fermion pairs.

# F   Anyon-vortex transmutation

In the main text, we focused on doped non-Abelian anyon fluids in the clean limit. The inclusion of quenched disorder changes the phase diagram of these systems in interesting ways. At sufficiently low doping $\delta$, disorder localizes charged anyons, preventing them from forming an itinerant phase. Past a critical dopant density $\delta_c$, excess anyons introduced into the system delocalize and transform the system into a superconductor or a correlated metal, thereby destroying the parent topological order. This modified phase diagram raises an important conceptual question: what is the fate of the localized anyons once we enter the itinerant phase? For Abelian anyon delocalization transitions, this question has been addressed in Ref. [54]. The goal of this appendix is to generalize the arguments in Ref. [54] to non-Abelian anyon delocalization transitions out of the Read-Rezayi states considered in this paper.

We begin with the Chern-Simons theory for the $\text{RR}_k$ topological order

$$L_{\text{RR}_k} = -\frac{k}{4\pi}\text{Tr}\left(c\,dc + \frac{2}{3}c^3\right) + \frac{k+1}{4\pi}\text{Tr}\,c\,d\,\text{Tr}\,c - \frac{1}{2\pi}\text{Tr}\,c\,dA + \text{CS}[A,g]. \tag{F.1}$$

As explained in the main text, the elementary non-Abelion $\sigma a$ with charge $e/(k+2)$ is sourced by a field $\Phi$ transforming in the $(1/2, 1)$ representation of $U(2)$. Finite-energy $\sigma a$ excitations can be introduced by adding a term in the Lagrangian $L_{(1/2,1)}[\Phi, c]$ which minimally couples the bosonic field $\Phi$ to the $U(2)$ gauge field $c$.

By dressing with powers of the electron operator $c$, the same anyon $\sigma a$ can also be represented as

$$(1/2, 1) = c^m \times A_m, \quad m \in \mathbb{Z}, \tag{F.2}$$

where $A_m$ is a non-Abelion carrying electric charge $-me + e/(k+2)$. The $m = 0$ case reduces to $A_0 = (1/2, 1)$ and the $m = 1$ case corresponds to the decomposition

$$(1/2, 1) = (k/2, k+2) \times (k/2 - 1/2, -k - 1), \tag{F.3}$$

where we used the fusion rules of $\text{RR}_k$ as well as the identification of the electron operator with $(k/2, k+2)$. This decomposition implies that finite-energy $\sigma a$ excitations can also be introduced by a different term in the Lagrangian $L_{(k/2-1/2,-k-1)}[f, A+c]$, where $f$ is a fermionic field transforming in the $(k/2 - 1/2, -k - 1)$ representation of $U(2)$. The coupling to the background $\text{spin}_{\mathbb{C}}$ connection $A$ encodes the dressing with an extra electron and requires that $f$ be a fermionic field.

Including both types of matter fields, we arrive at a Lagrangian for the doped $\text{RR}_k$ state

$$L = L_{\text{RR}_k} + L_{(1/2,1)}[\Phi, c] + L_{(k/2-1/2,-k-1)}[f, A+c]. \tag{F.4}$$

Each $\sigma a$ anyon is represented by a localized source of $\Phi$ or $f$, such that the total density $\rho_\Phi + \rho_f$ is equal to the density of localized anyons in the system.

Now let us go through a phase transition from the $\text{RR}_k$ state to one of the superconductors considered in the main text. In the paramagnetic case, $L_{\text{RR}_k}$ transforms into a Lagrangian for the $c_- = -1/2$ topological superconductor

$$\begin{aligned}L = \frac{2}{4\pi}\text{Tr}\left(cdc + \frac{2}{3}c^3\right) - \frac{1}{4\pi}\text{Tr}\,c\,d\,\text{Tr}\,c + \frac{1}{2\pi}\text{Tr}\,c\,dA + \text{CS}[A,g] \\ + L_{(1/2,1)}[\Phi, c] + L_{(k/2-1/2,-k-1)}[f, A+c].\end{aligned} \tag{F.5}$$

The equations of motion for the Abelian part of $c$ then take the form

$$\frac{2}{2\pi}\nabla \times A = \rho_\Phi - (k+1)\rho_f. \tag{F.6}$$

This equation implies that a single $\Phi$ excitation sources a $\pi$-flux of $\boldsymbol{A}$, while a single $f$ excitation sources a $-(k+1)\pi$ flux of $\boldsymbol{A}$. After restoring factors of $h$ and $e$, we then conclude that $\Phi$ and $f$ source superconducting vortices with vorticity $h/2e$ and $-(k+1)h/2e$ respectively. By dressing with higher powers of the electron operator, we see that the most general vortex originating from the $e/(k+2)$ anyon has vorticity $h/2e$ mod $(k+2)h/2e$.

Near the transition from the $RR_k$ plateau to the superconductor, only a small fraction of charge carriers delocalize. The argument above implies that localized charge carriers become localized vortices randomly distributed inside the superconductor. Since the superconductor prefers to have zero net vorticity energetically, we expect to find a random sprinkling of $h/2e$ and $-(k+1)h/2e$ vortices with a density ratio of $(k+1):1$. This is the non-Abelian version of the anomalous vortex glass state proposed in Ref. [54].

In the ferromagnetic case with even $k$, $L_{RR_k}$ transforms into a Lagrangian for the $c_- = 0$ charge-$ke$ superconductor

$$L = \frac{k}{2\pi}c_\uparrow dA + L[\Phi, c_\uparrow] + L[f, A - (k+1)c_\uparrow]. \tag{F.7}$$

The equations of motion for $c_\uparrow$ then give the constraint

$$\frac{k}{2\pi}\nabla \times \boldsymbol{A} = \rho_\Phi - (k+1)\rho_f. \tag{F.8}$$

As a result, a single $\Phi$ excitation sources a $h/ke$ vortex, while a single $f$ excitation sources a $-(k+1)h/ke$ vortex. By dressing with higher powers of the electron operator, we see that the most general vortex originating from the $e/(k+2)$ anyon has vorticity $h/ke$ mod $(k+2)h/ke$. Following the same logic as in the paramagnetic case, we conclude that the superconducting state close to the anyon delocalization transition is an anomalous vortex glass with random sprinkled $h/ke$ and $-(k+1)/ke$ vortices with a density ratio $(k+1):1$.

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
