# Peer review of "Doping lattice non-abelian quantum Hall states"

_SciPost Physics, doi:SciPost Phys. 19, 150 (2025)_

## Round 1 · Referee Report · Anonymous (Referee 2) · 2025-9-17

Disclosure of Generative AI use

The referee discloses that the following generative AI tools have been used in the preparation of this report:

check grammar.

Strengths

A very careful discussion of a complicated subject. Impressive technical skills.

Report

The manuscript by Shi et al. presents an impressive and comprehensive understanding of superconductivity emerging from doped non-Abelian topological orders. While anyon superconductivity is a well-studied subject, previous discussions have (to my knowledge) primarily focused on doping Abelian states, which is a much simpler situation than the one addressed here. This paper provides a careful analysis of emergent superconductivity in doped non-Abelian FQH states, and I believe it will serve as a valuable resource for further theoretical investigations. I recommend publication.

My only suggestion is that it would strengthen the paper if the authors could propose a clear “smoking-gun” experimental signature of the anyon superconductor. In particular, what experiment could unambiguously identify that the mechanism of the charge-2e superconductivity arises from doping charge-e/4 non-Abelian anyons, rather than form other mechanism?

Recommendation

Publish (easily meets expectations and criteria for this Journal; among top 50%)

  • validity: high
  • significance: good
  • originality: good
  • clarity: good
  • formatting: good
  • grammar: reasonable

Author:  Zhengyan Shi  on 2025-11-21  [id 6059]

(in reply to Report 2 on 2025-09-17)
Category:
answer to question

We thank the referee for their insightful comments. Following their suggestion, we have included an extended discussion of experimental signatures, supported by a new Appendix F. Indeed, in the absence of disorder, the charge-2e superconductor obtained from doping e/4 anyons in the Pfaffian state is smoothly connected to a regular p-ip BCS superconductor (i.e. all the universal low-energy properties match). However, when impurities are present, the superconductor in the vicinity of the Pfaffian-SC transition is a novel anomalous vortex glass phase, with randomly pinned vortices carrying vorticity h/2e or -3h/2e. The density of these vortices have a 3:1 ratio, such that the global vorticity vanishes. This pattern is difficult to obtain through conventional BCS/Kohn-Luttinger pairing and may be viewed as a smoking-gun for the anyon-driven mechanism. In the new discussion, we derive this phase and also comment on other experimental probes for more exotic higher-charge superconductors and higher-charge Fermi liquids obtained from doping Read-Rezayi states at filling k/(k+2).

---

## Round 1 · Referee Report · Anonymous (Referee 1) · 2025-9-17

Strengths

1- The paper deals with the problem of doping non-abelian quantum Hall states. Building on the work by some of the authors on doping abelian quantum Hall states, the paper presents a general framework for analyzing this conceptually and experimentally important problem.

2- The paper is written with good pedagogy in mind, carefully introducing important concepts in the main text, which is supported by a number of appendices.

3- In keeping with the aforementioned prior work, this work carefully considers the effect of the underlying lattice and the fractionalization of translation symmetry.

4- The paper tackles both Moore-Read states and general k Read-Rezayi states, showcasing distinct behavior between odd/even k's. The field theoretic derivation is complemented by a heuristic argument for the origin of this difference.

5- The paper also presents a heuristic charge-flux unbinding picture for the origin of superconductivity, but also presents the limit of this approach.

Weaknesses

1- Relative to the analysis of the field theory when the components fields go into insulators (such as bIQH), the argument around bosonic CFL seemed a little less tight. On the other hand, it is probably a consequence of Fermi liquids generally being more difficult to deal with than insulators, and I don't think any improvement is needed to make the paper suitable for a publication.

2- As the paper admits itself, the analysis depends strongly on mean-field assumptions about the nature of the ground states. While the assumptions look reasonable, it is sometime hard to keep track of how many assumptions are being made. For example, what happens when the bosons do not go into bIQH? How reasonable is that scenario? The paper touches on some of these questions (e.g. when discussing k->infinity limit), but it is a bit opaque in my opinion.

Report

The paper presents a compelling field theoretical analysis of doped non-abelian quantum Hall states. While the paper makes liberal use of mean-field assumptions, it serves as a good conceptual starting point for analyzing future analytical/numerical works on this topic, and I believe it should be published in SciPost physics.

Recommendation

Publish (surpasses expectations and criteria for this Journal; among top 10%)

  • validity: top
  • significance: high
  • originality: high
  • clarity: top
  • formatting: excellent
  • grammar: perfect

Author:  Zhengyan Shi  on 2025-11-21  [id 6060]

(in reply to Report 1 on 2025-09-17)
Category:
answer to question

We thank the referee for their insightful comments and criticisms. Let us address them in order.

Regarding the first criticism, we agree that it is more difficult to justify the formation of bosonic CFL phases that appear in our analysis. However, we remark that the bosonic CFL phase is in any case only used as an intermediate step towards obtaining a charge-ordered Fermi liquid ground state. In other words, we can view the bosonic CFL as a valid description at some intermediate energy scales which exposes an inevitable pairing instability at the lowest temperature. The paired state ends up being an ordinary Fermi liquid, over which we have much better analytical control.

Regarding the second criticism, we agree that there is some freedom in choosing the mean-field state. We have added a comment in the discussion section emphasizing that we are always choosing the simplest translation-preserving mean-field consistent with filling constraints. For example, when bosons are at an even-integer filling, we always assume that it forms the simplest invertible phase which is the boson IQH state. If we allow ourselves to consider more complicated states with symmetry-breaking and/or topological order, the number of possible phases proliferates and there is no definite conclusion to be drawn. It would be great to have a more rigorous justification for this Occam's razor approach in the future, either through numerics or some more sophisticated analytical methods.

---

## Editorial Decision

published